# MITIGATING POSITION BIAS IN TRANSFORMERS VIA LAYER-SPECIFIC POSITIONAL EMBEDDING SCALING

## ABSTRACT

Large language models (LLMs) still struggle with the "lost-in-the-middle" problem, where critical information located in the middle of long-context inputs is often underrepresented or lost. While existing methods attempt to address this by combining multi-scale rotary position embeddings (RoPE), they typically suffer from high latency or rely on suboptimal hand-crafted scaling. To overcome these limitations, we introduce a layer-specific positional embedding scaling (LPES) method that assigns distinct scaling factors to each layer. LPES achieves a more balanced attention distribution without fine-tuning model parameters or increasing inference delay. A specially designed genetic algorithm is employed to efficiently select the optimal scaling factors for each layer by incorporating Bézier curves to reduce the search space. Extensive experiments demonstrate that LPES effectively mitigates positional attention bias and delivers consistent improvements across multiple long-context benchmarks, yielding up to an 11.2% accuracy gain on the key-value retrieval dataset.

## 1 INTRODUCTION

Enabling large language models to effectively process long inputs is essential for supporting complex tasks such as long-text summarization (Feng et al., 2021; Zhang et al., 2021), code generation (Zheng et al., 2023; Liu et al., 2024a), and long-context question-answering (Li et al., 2024). Rotary position embeddings (RoPE) (Su et al., 2021), widely adopted in transformer-based LLMs, were designed to encode relative distances between input tokens and are expected to handle long inputs more effectively than absolute positional embeddings (Vaswani et al., 2017). However, as context length increases, RoPE-based LLMs still exhibit position bias, where the model fails to allocate appropriate attention across different positions in the input, even when the input length is within the model's pre-training range. A prominent instance of this problem is the well-known "lost-in-the-middle" phenomenon (Liu et al., 2024c), in which models disproportionately focus on the beginning and end of the context while relatively overlooking critical information in the middle. RoPE encodes relative positions through the superposition of sine and cosine functions with varying frequencies. The periodic and oscillatory nature of these functions causes inter-token dependencies to attenuate over longer distances (Chen et al., 2023b; Zhang et al., 2024), which can result in imbalanced attention distribution across the input sequence.

Several approaches have been proposed to address the position bias problem by combining multiple rotary position embeddings with different bases or scaling factors (Chen et al., 2023b; Zhang et al., 2024; Lin et al., 2024). Chen et al. (2023b) observed that RoPE with different bases induces attention troughs at specific positions, which impairs the model's ability to capture the corresponding content. To mitigate this, they introduced a method, named Attention Buckets, that combines multiple RoPE with different bases to achieve a more balanced attention distribution. Similarly, Lin et al. (2024) proposed an MoICE method that assigns multiple RoPE bases to each attention head and aggregates the outputs through a weighted sum. However, whether through varying bases or scaling factors, these methods require multiple forward passes during inference, each corresponding to a specific base or scaling factor, followed by ensembling the results. Although some operations can be parallelized, this process inevitably slows down inference and increases computational cost.

Varying bases across the entire model can be considered as a form of model-level ensembling, whereas applying multiple RoPE bases to individual attention heads functions as module-level ensembling. The former influences the model globally, limiting the flexibility for targeted adjustments,

Figure 1: Comparison of the proposed LPES with two representative existing methods. (a) Attention Buckets (Chen et al., 2023b) combines multiple RoPEs with different bases through model parallels. (b) MoICE (Lin et al., 2024) assigns multiple bases to each attention head. Unlike these existing methods which require multiple forward passes during inference, our LPES (c) achieves superior performance with a single forward pass, significantly reducing inference time.

while the latter operates at an overly fine-grained level (down to each individual attention head), making it challenging to determine suitable scaling factors and their weights for each head. In this study, we explore a control granularity that lies between these two extremes. Specifically, we propose applying different scaling factors at different layers, with all attention heads within a layer sharing the same factor. Most importantly, our method is designed to achieve or surpass the performance of existing methods with a single forward pass during inference, thereby eliminating the overhead for multiple passes.

Choosing an appropriate scaling factor for each layer is still a non-trivial problem. Let $L$ denote the number of layers in a transformer-based network, and $M$ the number of possible values for the scaling factors; the total number of combinations is $M^L$, which makes an exhaustive search computationally intractable. We attempted to use gradient backpropagation to determine the scaling factors for each layer; however, we observed poor convergence, and the resulting model performed worse than expected. This is because gradient descent tends to converge to a local optimum, whereas the problem is inherently combinatorial in nature. To overcome this, we leverage Bézier curves, which can define a smooth, continuous curve using a limited set of discrete control points. Denoting the number of control points by $C$, the search space is reduced to $(M \times L)^C$, with a detailed analysis provided in Appendix B. Our experiments show that a cubic Bézier curves (i.e., $C = 4$) can represent a wide variety of shapes and are sufficient to capture layer-specific scaling relationships. We also designed a genetic algorithm to solve this combinatorial optimization problem by constraining the search space to the Bézier curve. By combining the genetic algorithm with Bézier curves (see Figure 2), we can efficiently optimize layer-specific scaling factors, typically within 3 to 4 hours using only a few hundred examples (e.g., 200 instances) on four H100 GPUs. In long-text tasks, our method introduces no additional inference latency while delivering superior performance over existing approaches, without requiring fine-tuning of the LLM parameters.

The contributions of this study can be summarized as follows:

- We propose a layer-specific positional embedding scaling method, termed LPES, which effectively mitigates the position bias problem without incurring additional inference latency. LPES achieves significant speedups, 2.42x faster than MoICE (Lin et al., 2024) and 1.45x faster than Ms-PoE (Zhang et al., 2024), while also improving the model's ability to handle long-context tasks.
- We introduce an efficient genetic search algorithm in which the search space is constrained by Bézier curves, enabling rapid optimization of layer-specific scaling factors using only a small set of examples (typically a few hundred examples only).
- Extensive experiments on multiple benchmark datasets demonstrate that our method preserves the model's general capabilities while producing a more balanced attention distribution without costly fine-tuning, making it broadly applicable across different models and tasks.

## 2 RELATED WORK

Assigning attention based on the relevance of information rather than its position, is a cornerstone of transformer-based LLMs. For instance, retrieval-augmented generation (RAG) has proven effective in incorporating up-to-date knowledge, reducing hallucinations, and improving response quality (Gao et al., 2023; Wang et al., 2024). The success of RAG lies in its ability to enhance generation whenever query-relevant documents are available. In practice, multiple documents are typically re-

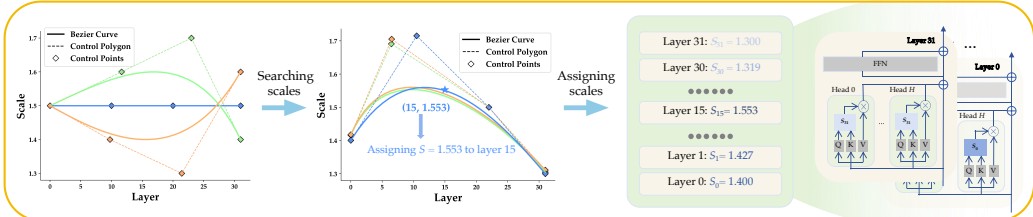

Figure 2: Illustration of the proposed layer-specific positional embedding scaling (LPES) method. Left: Bézier curves can represent a wide variety of shapes. Middle: An optimized Bézier curve found by our search algorithm, which defines a smooth, continuous curve using a limited set of discrete control points. Right: The relationship between the scaling factors and the optimized Bézier curve, and their application within the attention mechanism of a transformer-based network.

trieved, and it is impossible to determine in advance which ones are most useful or to place them where attention is concentrated. Consequently, it is crucial for LLMs to maintain a balanced attention distribution across all input positions. Nevertheless, even when the input length remains within the model's pre-training range, RoPE-based LLMs still exhibit position bias (Chen et al., 2023b; Zhang et al., 2024). Addressing this bias and achieving a more balanced attention distribution as input lengths increase has attracted considerable research interest (He et al., 2024; An et al., 2024; Peysakhovich & Lerer, 2023; Hsieh et al., 2024; Chen et al., 2023b; Zhang et al., 2024; Lin et al., 2024), with existing approaches falling broadly into four categories: fine-tuning, repositioning, attention score reassignment, and multi-scale positional embeddings.

In fine-tuning approaches (He et al., 2024; An et al., 2024), a domain-specific dataset needs to be constructed, and instruction fine-tuning is then applied to guide the model toward focusing more on the relevant parts of the context. However, those approaches demands substantial human effort to curate a large volume of training samples and entails significant computational costs during the fine-tuning process. In many cases, particularly when LLMs have already been deployed in real-world applications, it is preferable not to update their parameters in order to minimize potential impacts.

The repositioning method (Peysakhovich & Lerer, 2023) begins by computing attention scores for each document in the input. Documents are then reordered based on these scores, with the highest-scoring documents placed closer to the query. After this initial reordering, the attention scores are recomputed and the document positions are updated accordingly. This iterative process continues several times. Nevertheless, such repeated recomputation and reordering substantially increase the inference-time cost. In contrast, Hsieh et al. (2024) collect attention scores for all input positions from a set of examples and rescale them for each attention head in order to distribute attention as evenly as possible across positions. However, this form of attention score reassignment may lead to instability in the generation, particularly when the calibration intervenes at early layers.

Chen et al. (2023b) identified that RoPE with different bases can produce attention troughs at specific positions, thereby impairing the model's ability to capture the relevant content. To address this, their "attention buckets" method integrates multiple RoPE bases through model-parallel inference to achieve a more uniform attention distribution. Zhang et al. (2024) suggested that the long-term decay in attention may contribute to the position bias, and proposed Ms-PoE that assigns distinct scaling factors to attention heads based on their relative sensitivity to positional information. Building on the work of (Chen et al., 2023b), MoICE employs gradient descent to learn the weights for combining results with different bases (Lin et al., 2024). However, a major limitation of these approaches is their high computational cost and inference latency. Specifically, attention buckets requires multiple forward passes, while both Ms-PoE and MoICE require repeated attention computations to integrate multi-scale RoPE information. These methods also rely on heuristic rules or hand-crafted configurations to select bases or scaling factors. By contrast, our method not only achieves superior performance with a single forward pass at inference but also introduces a search algorithm that can efficiently determine the optimal scaling factors using only a few hundred examples.

## 3 METHOD

The objective of this study is to achieve a more balanced attention distribution across the entire input sequence by assigning distinct RoPE scaling factors to each transformer layer without incurring additional inference costs. Previous research has shown that different network layers exhibit

attention biases toward certain input positions, and these biases can be modulated by adjusting the RoPE scaling factors, which were originally introduced to extend the network's context window length (Vig & Belinkov, 2019; Lis et al., 2022; Zhai et al., 2023). Unfortunately, our preliminary experiments indicate that position biases cannot be substantially mitigated by simply assuming that scaling factors vary linearly with layer depth, which significantly enlarges the search space of the problem. Fortunately, Bézier curves provide an appealing alternative, as they can represent a wide variety of shapes using only a small set of discrete control points (see Figure 2). However, selecting the optimal scaling factors remains a combinatorial optimization problem, even with the help of Bézier curves. We tried to use an end-to-end gradient descent algorithm to determine these factors, but it performed poorly (see Appendix C), as the algorithm tend to converge to suboptimal local minima. In the following, we first present the formal problem formulation and then introduce a specially designed genetic algorithm to solve this problem in detail.

### 3.1 PROBLEM DEFINITION

In this study, we focus on improvements based on rotary position embeddings (RoPE), which have proven superior to their predecessor, absolute positional embeddings (Clark et al., 2020; Lan et al., 2019), due to its capability to enable language models to extrapolate effectively to longer sequences (Su et al., 2021). With RoPE, the relative distance between tokens can be computed simply through the inner product of their vector representations:

$$\langle f(\boldsymbol{q}, i), f(\boldsymbol{k}, j) \rangle = \boldsymbol{q}^{\mathrm{T}} R(i-j) \boldsymbol{k} \tag{1}$$

where $f(\boldsymbol{x}, i)$ denotes the RoPE operation, which applies a position-dependent rotation at position $i$ to the query $\boldsymbol{q}$, and $f(\boldsymbol{k}, j)$ represents the RoPE-rotated key at position $j$. The notation $\langle \cdot, \cdot \rangle$ denotes the inner product between the two position-aware vectors, and $R(\Delta)$ is the rotation corresponding to the relative offset $\Delta = i - j$. This equation shows that the inner product depends only on the vectors $\boldsymbol{q}, \boldsymbol{k}$ and the relative distance between them. Chen et al. (2023a) further show that the context window length can be extended by applying a scaling factor $s$ to the position as follows:

$$f'(\boldsymbol{x}, i) = f(\boldsymbol{x}, i/s) \tag{2}$$

By reducing the interval between two positions through this scaling, the context window length is extended by a factor of $s$ in theory. We further show that the scaling factors can mitigate long-term decay and produce diverse attention patterns (see Appendix A). Our objective is to search for a unique scaling factor $s$ for each layer to mitigate the position bias problem.

As previously mentioned, determining the optimal scaling factor for each layer by a brute-force approach is intractable, since each factor can take many possible values and transformers typically consist of tens of layers. To address this challenge, we represent the scaling factors for all layers using a single Bézier curve. As illustrated in Figure 2, a Bézier curve here can be viewed as a smooth curve that connects all the scaling factors in a two-dimensional plane. In this way, the problem of selecting scaling factors for all layers is transformed into the problem of searching for an appropriate Bézier curve. Fortunately, Bézier curves can model a wide variety of shapes using only a small set of discrete control points, which significantly reduces the search space. A Bézier curve of degree $d$ which has $d + 1$ control points is defined as follows (Mortenson, 1999):

$$B(t) = \sum_{k=0}^{d} b_k^d(t) P_k, \quad 0 \le t \le 1. \tag{3}$$

where the variable $t$ represents the parametric coordinate that controls a point's position along the curve, $P_k$ are the control points for the curve, and $b_k^d$ are the Bernstein basis polynomials, which are defined as:

$$b_k^d(t) = \frac{d!}{k!(d-k)!} t^k (1-t)^{d-k}, \quad k = 0, \dots, d. \tag{4}$$

Once a Bézier curve is determined, the scaling factor $s_h$ for layer $h$ can be computed as follows:

$$s_h = \mathrm{proj}_y \left[ B(t(x_h)) \right] \tag{5}$$

where the notation $\mathrm{proj}_y[\cdot]$ denotes the operation of extracting the $y$-coordinate of a two-dimensional point. The function $t(\cdot)$ maps $x_h$ to the corresponding parameter $t$ (see Appendix D), where $x_h$

represents the position of layer $h$ within the evenly spaced $x$-coordinates defined by the minimum and maximum values of the control points. The value of $x_h$ can be computed by:

$$x_h = P_0^x + \frac{P_d^x - P_0^x}{L - 1} \cdot h, \quad h = 0, \ldots, L - 1. \tag{6}$$

where $L$ denotes the number of layers in a network, and $P_t^x$ is the $x$-coordinates of the $t$-th control point for the Bézier curve.

Given a training dataset $\mathcal{D} = \{(x_i, y_i)\}_{i=1}^N$ consisting of $N$ examples, where $x_i$ is an input to the large language model and $y_i$ is the corresponding ground-truth output, our goal is to maximize the following function:

$$\mathcal{L}_\mathcal{D}(\boldsymbol{\theta}) = \frac{1}{N} \sum_{i=1}^N \mathbb{I}\{\text{LLM}(x_i, \boldsymbol{\theta}) \simeq y_i\} \tag{7}$$

where $\boldsymbol{\theta} = (P_0, \ldots, P_d)$ denotes the set of control points defining a Bézier curve of degree $d$ (each control point $P_k$ is a two-dimensional point), $\text{LLM}(x_i, \boldsymbol{\theta})$ denotes the output of a language model given input $x_i$, with all scaling factors determined according to Equation (5) based on the Bézier curve specified by $\boldsymbol{\theta}$, and $\mathbb{I}\{\cdot\}$ is an indicator function with binary output 0 or 1. We constructed the training dataset such that the content containing information useful for generating correct answers appears at varying positions within the input, thereby encouraging the model to distribute its attention more evenly across the entire input.

## 3.2 Optimization Algorithm

We can regard $\boldsymbol{\theta} = (P_0, \ldots, P_d)$ as a set of newly introduced hyper-parameters that influence the behavior of an LLM. Each $P_k$ is a two-dimensional vector whose $x$- and $y$-coordinates can take multiple different values. Even though Bézier curves of degree $d = 3$ which has $d + 1 = 4$ control points, are capable of representing a wide variety of curves, selecting suitable control points constitutes a combinatorial optimization problem. Due to the high complexity of the search space, a brute-force approach for determining the scaling factors across layers is intractable; instead, we employ a genetic algorithm to optimize the control points of the Bézier curves.

In our genetic algorithm, each individual is represented as $(P_0^x, P_0^y, \ldots, P_d^x, P_d^y)$, where $P_k^x$ and $P_k^y$ denote the $x$- and $y$-coordinates values of the $k$-th control point, and each individual corresponds to a specific Bézier curve. The initial population is constructed as follows. First, we initialize an individual in which $k$-th control point is generated by:

$$(k(L-1)/d, 1.5), \ k \in \{0, \ldots, d\} \tag{8}$$

where $L$ is the number of layers in a network. Based on the empirical results reported by Zhang et al. (2024), we set the $y$-coordinate values of all control points to 1.5. Subsequently, the remaining individuals are generated by applying a mutation operator (described below) to this initial individual until the population reaches the predefined size.

The fitness of an individual $\boldsymbol{\theta} = (P_0^x, P_0^y, \ldots, P_d^x, P_d^y)$ is evaluated by configuring the layer-wise scaling factors of an LLM according to $\boldsymbol{\theta}$, running the LLM on a dataset $\mathcal{D}$, and calculating the resulting score $\mathcal{L}_\mathcal{D}(\boldsymbol{\theta})$ as defined in Equation (7). When constructing the training dataset, we deliberately vary the position of relevant context within the input, which can generally be categorized into three types: the query-relevant content appears at the beginning, middle, or end of the input sequence. We denote these three corresponding sub-datasets as $\mathcal{D}_\text{B}$, $\mathcal{D}_\text{M}$, and $\mathcal{D}_\text{E}$, respectively. Considering that original LLMs tend to allocate attention unevenly across different positions, we introduce three weights to reflect the relative importance of these sub-datasets when optimizing the model's scaling factors. The final fitness of an individual is then computed as $\lambda_\text{B}\mathcal{L}_{\mathcal{D}_\text{B}}(\boldsymbol{\theta}) + \lambda_\text{M}\mathcal{L}_{\mathcal{D}_\text{M}}(\boldsymbol{\theta}) + \lambda_\text{E}\mathcal{L}_{\mathcal{D}_\text{E}}(\boldsymbol{\theta})$, where $\lambda_\text{B} \geq 0$, $\lambda_\text{M} \geq 0$, $\lambda_\text{E} \geq 0$, and $\lambda_\text{B} + \lambda_\text{M} + \lambda_\text{E} = 1$.

The crossover operator is performed by randomly selecting a pair of individuals with relatively high fitness scores as parents, choosing a single crossover point at random, and exchanging the segments beyond this point between the parents. This process produces two offspring, from which we retain only the one with the higher fitness.

The mutation operator modifies $P^x$ and $P^y$ within a specified range to prevent excessive variations in the resulting curve, as shown in Equation (9). Let $M_x$ and $M_y$ denote the maximum allowable

change for the $x$- and $y$-coordinate, respectively. After mutation, the $k$-th control point $(\hat{P}_k^x, \hat{P}_k^y)$ of an individual must remain within the following range:

$$\hat{P}_k^x \in \begin{cases} [\max(0, P_k^x - M_x), \ \min(P_{k+1}^x, P_k^x + M_x)] & \text{if } k = 0, \\ [\max(P_{k-1}^x, P_k^x - M_x), \ \min(P_{k+1}^x, P_k^x + M_x)] & \text{if } 0 < k < d, \\ [\max(P_{k-1}^x, P_k^x - M_x), \ \min(P_k^x + M_x, L-1)] & \text{if } k = d \end{cases} \tag{9}$$

$$\hat{P}_k^y \in \left\{ [\max(1, P_k^y - M_y), \ \min(P_k^y + M_y, 2)] \quad \text{if } 0 \le k \le d \right.$$

To ensure the smoothness of the curve and prevent undesirable abrupt changes in the scaling factor (Ding et al., 2024), the $x$-coordinate values of all control points must increase monotonically. The following condition should therefore be satisfied when performing either crossover or mutation operations. Let $P_i^x$ and $P_j^x$ denote the $x$-coordinates of the $i$-th and $j$-th control points, respectively. Their relationship is required to satisfy:

$$0 \le P_i^x < P_j^x \le n - 1 \quad \text{if} \quad i < j \tag{10}$$

Offspring that fail to meet the above condition are discarded, and the crossover or mutation process is repeated until the condition is satisfied.

Starting with the initial population, individuals are selected based on their fitness, followed by the application of the crossover and mutation operators. This process is repeated iteratively until the maximum number of generations is reached. The complete process is summarized in Algorithm 1.

---

**Algorithm 1** Layer-specific scaling factor search algorithm

---

**Input:** an LLM $\mathcal{M}$, a dataset $\mathcal{D}$, population size $N_{\text{ps}}$, the number of offspring generated by crossover $N_{\text{cr}}$,
      the number of mutated individuals $N_{\text{mu}}$, and maximum number of generations $T$.

1:   $\mathcal{S}_0$ = Initial-Population-Generation($\mathcal{D}$, $N_{\text{ps}}$); // Randomly generate the initial population.
2: **for** $i = 1$ to $T$ **do**
3:      Evaluate-Fitness($\mathcal{S}_{i-1}$, $\mathcal{M}$, $\mathcal{D}$); // Evaluate the fitness of all individuals in the population.
4:      $\mathcal{S}_{\text{pa}}$ = Select-Parents($\mathcal{S}_{i-1}$); // Select the parent pool according to fitness values.
5:      $\mathcal{S}_{\text{cr}}$ = Crossover-Operator($\mathcal{S}_{\text{pa}}$, $N_{\text{cr}}$); // Produce offspring using the crossover operator.
6:      $\mathcal{S}_{\text{mu}}$ = Mutation-Operator($\mathcal{S}_{\text{pa}}$, $N_{\text{mu}}$); // Generate offspring using the mutation operator.
7:      $\mathcal{S}_i = \mathcal{S}_{\text{pa}} \cup \mathcal{S}_{\text{cr}} \cup \mathcal{S}_{\text{mu}}$; // Merge the individuals to form the next generation's population.
8: **end for**
9: Return the individual with the highest fitness in $\mathcal{S}_T$.

---

## 4   EXPERIMENT

We conducted four sets of experiments. The first evaluates the performance of LPES when relevant information is placed at different positions in the input, examining how effectively LPES promotes a more balanced attention distribution across the input sequence. The second experiment assesses the performance of LPES on both open-ended and closed-ended benchmark datasets, in comparison with baseline methods. The third investigates the impact of the proposed method on the general capabilities of LLMs, as well as its inference efficiency relative to existing approaches. The final experiment examines how the choice of hyperparameter values influences the performance of LPES.

### 4.1   MODELS, DATASETS, AND BASELINES

**Base Models:** We selected four representative RoPE-based LLMs for our experiments: Vicuna-7B-v1.5 (Chiang et al., 2023), LLaMA-2-7B-chat (Touvron et al., 2023), and StableBeluga-7B (Mahan et al., 2023), each with a 4k-token context window, as well as Qwen2.5-7B (Yang et al., 2024), which supports a 130k-token context window.

**Benchmark Datasets:** MDQA (Liu et al., 2024b) is a widely-used multi-document question answering dataset. The key-value retrieval dataset (Liu et al., 2024b) consists of key–value pairs in which both keys and values are universally unique identifiers (UUIDs), making it particularly suitable for evaluating a model's ability to extract relevant information. ZeroSCROLLS (Shaham et al., 2023) comprises multiple open-ended long-text task datasets, with the specific sub-datasets and evaluation metrics summarized in Table 14. For closed-ended tasks, we adopt L-Eval (An et al., 2023) to

| Models | Methods | 0% | 25% | 50% | 75% | 100% | Average | 0% | 20% | 40% | 60% | 80% | 100% | Average |
|---|---|---|---|---|---|---|---|---|---|---|---|---|---|---|
| | | | | MDQA | | | | | | Key-Value Retrieval | | | | |
| Vicuna-7B-v1.5 | Baseline | 70.4 | 58.0 | 55.4 | 55.4 | 60.4 | 59.9 | 95.2 | 71.6 | 81.0 | 79.0 | 77.4 | 73.4 | 80.9 |
| | Positional Interpolation | 71.2 | 59.6 | 58.8 | 56.4 | 56.2 | 60.4 | 98.6 | 92.8 | 83.8 | 90.0 | 85.8 | 83.0 | 89.0 |
| | Attention Buckets | **72.6** | 61.4 | 60.6 | 60.8 | 59.6 | 63.0 | **100** | **94.6** | 88.6 | 91.6 | 87.6 | 65.8 | 88.0 |
| | Ms-PoE | **72.6** | 61.4 | 61.8 | **62.0** | 59.0 | 63.5 | 95.2 | 63.2 | 84.8 | 91.6 | 87.4 | 77.8 | 83.3 |
| | MoICE | 71.6 | 61.2 | 60.6 | 60.8 | **62.4** | 63.3 | **100** | 93.2 | **90.2** | 87.4 | 89.4 | 70.0 | 88.4 |
| | LPES (Ours) | 71.4 | **62.2** | 62.0 | 61.0 | 61.6 | **63.6** | 99.4 | 92.8 | 87.8 | **93.6** | 90.4 | 88.8 | **92.1** |
| StableBeluga-7B | Baseline | 67.8 | 59.2 | 59.6 | 59.4 | 68.2 | 62.8 | 90.2 | 34.2 | 44.0 | 16.6 | 59.8 | 79.4 | 54.0 |
| | Positional Interpolation | **69.6** | 58.6 | 58.2 | 60.0 | 65.4 | 62.4 | 95.2 | 53.6 | 31.8 | 28.6 | 61.6 | 83.6 | 59.1 |
| | Attention Buckets | 69.2 | 59.0 | 59.8 | 59.2 | 67.4 | 63.0 | **100** | 79.8 | 54.4 | 58.2 | 68.4 | 89.2 | 75.6 |
| | Ms-PoE | 68.4 | 57.0 | 60.2 | **61.0** | 68.4 | 63.0 | 90.2 | 27.2 | 27.6 | **59.4** | 70.4 | 89.0 | 60.6 |
| | MoICE | 67.4 | 60.0 | 60.2 | 60.0 | **68.6** | 63.2 | 99.8 | 71.2 | 52.2 | 54.8 | **74.4** | 91.4 | 74.0 |
| | LPES (Ours) | 68.8 | **60.0** | 60.8 | 61.0 | 68.2 | **64.5** | 99.2 | **82.4** | 57.2 | 56.2 | 70.4 | **89.6** | **75.8** |
| Qwen2.5-7B | Baseline | 69.4 | 61.0 | 62.6 | 58.6 | 63.6 | 63.0 | 99.8 | 88.6 | 92.6 | 90.6 | 99.0 | 99.2 | 95.0 |
| | Positional Interpolation | 68.6 | 62.0 | 62.2 | 58.4 | 64.0 | 63.0 | **100** | 93.2 | 91.2 | 88.6 | 98.6 | 99.0 | 95.1 |
| | Attention Buckets | 69.6 | 62.2 | 63.0 | 60.2 | 62.0 | 63.4 | **100** | 89.2 | 91.4 | 91.6 | 98.2 | 99.2 | 94.9 |
| | Ms-PoE | – | – | – | – | – | – | – | – | – | – | – | – | – |
| | MoICE | 68.4 | 61.2 | 63.0 | 61.0 | 63.8 | 63.5 | 99.8 | 88.0 | 92.6 | 91.6 | 99.0 | **99.4** | 95.1 |
| | LPES (Ours) | 69.6 | **64.8** | **69.2** | **63.0** | 65.4 | **66.4** | 99.8 | **97.4** | **93.2** | **94.0** | **99.2** | 99.2 | **97.1** |

Table 1: Performance (Accuracy) of LPES when relevant information is located at different positions (e.g., $50\%$ indicates that the relevant document is positioned in the middle), compared with baseline methods. LPES outperforms all baselines on average across multiple base models and datasets, demonstrating its effectiveness in mitigating positional bias.

assess model performance, with the description detailed in Table 13 (Appendix F). Finally, MMLU (Hendrycks et al., 2020) and C-Eval (Huang et al., 2023), which cover a broad range of general tasks, are employed to evaluate the overall generalization capability of the models.

**Baseline Methods:** Positional Interpolation (PI) uses a layer-agnostic scaling factor, which is the mean of the searched layer-wise scaling factors (Chen et al., 2023a). Attention Buckets performs multiple forward passes, each using a different RoPE base, and then aggregates the information from these passes (Chen et al., 2023b). Ms-PoE dynamically assigns scaling factors ranging from $1.2$ to $1.8$ to attention heads based on their sensitivity to relevant information (Zhang et al., 2024). Building on the work of Chen et al. (2023b), MoICE computes attention scores using seven different RoPE bases and then performs a weighted sum of these scores using learned weights (Lin et al., 2024).

**Experimental Setup:** For LLMs with a 4k-token context window, we use 10 documents from the MDQA dataset or 50 key–value pairs from the key-value retrieval dataset as context. To evaluate the effectiveness of mitigating positional bias in longer contexts, we provide Qwen2.5-7B with 20 MDQA documents or 150 key–value pairs and assess the model's accuracy when the ground-truth information appears at different positions within the context. For ZeroSCROLLS and L-Eval, the context window is set to 3,584 tokens, with a maximum of 512 decoded tokens (Tables 2 and 3). In addition to experiments conducted under the 4K context setting, we also report the performance of LPES under a 16K context window in Appendix I. In the optimization algorithm, we set $\lambda_{\text{B}}$, $\lambda_{\text{M}}$, and $\lambda_{\text{E}}$ to 0.2, 0.3, and 0.5, respectively, with their settings further examined in Appendix G. To determine the layer-wise scaling factors, we sample 200 examples from either the MDQA or key–value retrieval datasets to search the control points of cubic Bézier. The performance of LLMs with the optimized scaling factors is then evaluated on 500 held-out samples per dataset. To assess the generalizability of our LPES, the scaling factors learned from MDQA are also applied to ZeroSCROLLS and L-Eval. Additionally, the MMLU and C-Eval benchmarks are used to evaluate the effect of layer-specific scaling on the model's overall generalization capabilities.

## 4.2 RESULTS

*Layer-specific positional embedding scaling greatly mitigates position bias.* As shown in Table 1, our method consistently improves performance across all positions, whereas baselines like Positional Interpolation and Ms-PoE suffer from performance degradation when relevant information is at certain positions. Moreover, our LPES provides substantial gains on the key-value retrieval dataset, with an average increase of 11.2 observed for the Vicuna. We evaluate the transferability of the scaling factors derived from MDQA dataset on ZeroSCROLLS and L-Eval benchmarks. The results in Tables 2 and 3 demonstrate that our method is effective across different models and tasks, and for a given LLM, the optimized scaling factors generalize well to diverse tasks. Furthermore, the results on longer context windows and larger model scales (detailed in Appendix I) further validate

the applicability of LPES. Additionally, our method preserves the model's general capabilities with minimal interference, as shown in Table 4.

| Model | Method | GovRpt | Qasper | SumScrFd | Qmsum | NarrQA | Squality | SpcDgst | Average |
|-------|--------|--------|--------|----------|-------|--------|----------|---------|---------|
| Vicuna-7B-v1.5 | Baseline | 18.44 | 22.82 | 18.42 | 14.50 | 10.98 | 16.56 | 21.39 | 16.91 |
| | MoICE | **22.29** | 32.34 | 13.31 | 14.79 | **13.61** | 16.22 | **22.60** | 19.30 |
| | LPES (Ours) | 21.47 | **33.37** | 14.39 | **15.53** | 11.52 | **16.91** | 22.24 | **19.35** |
| LLaMA-2-7B-chat | Baseline | 18.00 | 13.48 | 13.73 | 14.29 | 10.28 | 15.94 | 49.72 | 19.35 |
| | MoICE | **19.62** | 15.10 | **14.69** | 14.79 | 10.25 | 16.80 | 50.22 | 20.21 |
| | LPES (Ours) | 18.20 | **15.23** | 13.99 | **15.04** | 14.93 | 17.37 | 52.28 | **21.01** |
| StableBeluga-7B | Baseline | 14.88 | 26.89 | 12.09 | 14.24 | 10.73 | 15.05 | 48.50 | 20.34 |
| | MoICE | 18.14 | **36.89** | 14.35 | 15.76 | 7.990 | 15.97 | 44.50 | 21.94 |
| | LPES (Ours) | **18.98** | 34.19 | 13.06 | 15.46 | 9.910 | 16.65 | 46.61 | **22.12** |
| Qwen2.5-7B | Baseline | 24.76 | 22.92 | 14.69 | 16.25 | 9.780 | 14.85 | 53.66 | 22.42 |
| | MoICE | 25.56 | 23.51 | 15.12 | 23.19 | 10.64 | **16.92** | 53.81 | 24.11 |
| | LPES (Ours) | **27.56** | **23.91** | **16.18** | 23.19 | **11.97** | 14.92 | 53.81 | **25.51** |

Table 2: Results of our method on various open-ended datasets compared with the baselines. Our LPES improves performance across multiple models on seven different long-text tasks, demonstrating its effectiveness in enhancing the model's ability to leverage contextual information.

| Model | Method | Coursera | QuALITY | TOEFL | SFiction | Average |
|-------|--------|----------|---------|-------|----------|---------|
| Vicuna-7B-v1.5 | Baseline | 37.21 | 38.12 | 38.00 | 57.90 | 42.81 |
| | MoICE | **46.65** | **43.71** | 39.33 | 57.20 | **46.72** |
| | LPES (Ours) | 40.41 | 42.57 | **40.67** | **58.20** | 45.46 |
| LLaMA-2-7B-chat | Baseline | 34.89 | 37.62 | 55.00 | 60.93 | 47.11 |
| | MoICE | **42.50** | 42.08 | 56.13 | 64.84 | 50.72 |
| | LPES (Ours) | 37.50 | **42.16** | **63.00** | 63.50 | **51.52** |
| Qwen2.5-7B | Baseline | 45.47 | 62.43 | 66.00 | 60.87 | 58.69 |
| | MoICE | 48.13 | 64.28 | 67.33 | 66.00 | 61.44 |
| | LPES (Ours) | **48.51** | **66.43** | **69.28** | **66.42** | **62.66** |

Table 3: Results of our method on four closed-ended long-text tasks compared with the baselines. Our LPES consistently enhances performance on four datasets across all models.

*LPES yields a more balanced attention distribution without incurring additional computational cost during inference.* Both Ms-PoE and MoICE are sample-dependent, requiring the scaling parameters to be determined for each individual input. Consequently, the scaling factors cannot be precomputed. Ms-PoE (Zhang et al., 2024) requires real-time computation of each attention head's sensitivity to relevant information, which entails performing attention calculations twice. MoICE, on the other hand, necessitates parallel attention computations across all seven modules, while the router computation is executed serially alongside the attention operations. To demonstrate the advantage in inference efficiency, we sample 500 examples from the MDQA dataset and report the average inference time of Vicuna on a single H100 GPU. For a fair comparison, FlashAttention-2 (Dao, 2023) was used as the attention backend for all methods. As shown in Table 5, the inference time per sample is 1.03 seconds for Ms-PoE, 1.72 seconds for MoICE, and approximately 0.71 seconds for our method, making LPES roughly 1.45x faster than Ms-PoE and 2.42x faster than MoICE.

| Model | Method | MMLU | C-Eval |
|-------|--------|------|--------|
| Vicuna-7B-v1.5 | Baseline | 49.90 | 49.42 |
| | LPES (Ours) | 49.00 | 49.33 |
| StableBeluga-7B | Baseline | 51.50 | 34.78 |
| | LPES (Ours) | 51.30 | 34.63 |

Table 4: General capability of models equipped with LPES on MMLU and C-Eval datasets.

| Method | Inference time per sample (s) |
|--------|-------------------------------|
| Baseline | 0.71 |
| Attention Buckets | 3.38 |
| Ms-PoE | 1.03 |
| MoICE | 1.72 |
| LPES (Ours) | **0.71** |

Table 5: Comparison of inference efficiency between LPES and baseline methods.

## 4.3 Impact of Hyper-Parameters

In this section, we present three sets of experiments using Vicuna-v1.5 on the MDQA dataset to demonstrate that Bézier curves more effectively determine layer-specific scaling factors compared to alternative curves. We further examine the impact of the number of control points on both convergence quality and speed. Compared to brute-force search, modeling the search space with Bézier curves enables rapid convergence to high-performing solutions within a limited time, and the results indicate that performance is largely insensitive to the number of control points employed. Additionally, by sampling different search sets to determine the scaling factors, we observe consistently stable performance, further confirming the robustness of our search algorithm.

### 4.3.1 The Impact of Curve Type

Bézier curves provide a compact, low-dimensional parameterization capable of approximating a wide variety of curve shapes (Nuntawisuttiwong & Dejdumrong, 2021). To demonstrate the advantages of Bézier curve modeling, we further employ two alternative approaches: linear interpolation between control points and step-function modeling based on control points. Although these alternatives differ in their curve formulations, they also serve as layer-specific scaling strategies within our framework. While linear interpolation offers slightly higher computational efficiency, we ultimately adopt Bézier curves due to their superior performance. As shown in Table 6, Bézier curves outperform other curve-fitting methods, and the minor additional cost required to determine the scaling factors is fully offset by the inference-time performance gains. Furthermore, when evaluated on deeper models and finer-grained positional segments, Bézier curves consistently yield improvements across all positions compared with linear interpolation, as detailed in Appendix H.

| Method | 0% | 25% | 50% | 75% | 100% | Average |
|---|---|---|---|---|---|---|
| Baseline | 70.4 | 58.0 | 55.4 | 55.4 | 60.4 | 59.92 |
| LPES (Linear interpolation) | **71.8** | 61.0 | **62.2** | 60.0 | 60.6 | 63.12 |
| LPES (Step function) | 71.6 | 60.2 | 59.4 | 59.2 | 60.4 | 62.16 |
| LPES (Bézier curve) | 71.4 | **62.2** | 62.0 | **61.0** | **61.6** | **63.64** |

Table 6: Performance comparison of different curve types for determining layer-wise scaling factors. Bézier curves achieve superior performance.

### 4.3.2 Effect of the Number of Control Points

We set the maximum number of iterations to 20 and, while keeping all other experimental settings unchanged, vary the number of control points to evaluate performance and convergence speed, where performance is measured by the mean and variance of accuracy across different document positions. As the number of control points increases, the Bézier curve fitting becomes more precise, improving the likelihood of identifying an optimal combination of scaling factors. However, a larger number of control points also enlarges the search space, which slows convergence.

As shown in Table 7, using four control points provides a favorable trade-off between performance and convergence speed. In contrast, brute-force search shows little tendency to converge within the limited number of iterations, further highlighting the efficiency of our constrained genetic algorithm.

| Control Points | Accuracy (Std) | Epochs to Convergence ($\leq 20$) |
|---|---|---|
| Baseline | 59.9 ($\pm 5.56$) | $--$ |
| Brute-Force | 60.2 ($\pm 4.69$) | 20 |
| 2 | 60.6 ($\pm 4.55$) | 3 |
| 3 | 62.2 ($\pm 3.97$) | 5 |
| 4 | 63.6 ($\pm 3.90$) | 9 |
| 5 | **63.8 ($\pm 3.87$)** | 16 |

Table 7: Performance with varying numbers of control points. Increasing the number of control points improves accuracy and reduces positional bias, although convergence slows as the computational cost of optimization increases. Overall, the experimental results indicate that performance is insensitive to the number of control points used.

### 4.3.3 ROBUSTNESS OF THE SEARCH ALGORITHM

In this section, we evaluate the robustness of the scaling factors with respect to variations in the search dataset. On the MDQA dataset, we use Vicuna-1.5-7B and randomly sample 200 training instances as the search set for each run. Across five independent runs, the average performance is 63.68 with a sample variance of only 0.027, demonstrating that our method remains highly stable under different search subsets. Overall, our approach consistently outperforms prior methods, highlighting both the stability and robustness of the proposed search algorithm.

| Method | 0% | 25% | 50% | 75% | 100% | Average |
|---|---|---|---|---|---|---|
| Baseline | 70.4 | 58.0 | 55.4 | 55.4 | 60.4 | 59.9 |
| Attention Buckets | **72.6** | 61.4 | 60.6 | 60.8 | 59.6 | 63.0 |
| Ms-PoE | **72.6** | 61.4 | 61.8 | 62.0 | 59.0 | 63.5 |
| MoICE | 71.6 | 61.2 | 60.6 | 60.8 | **62.4** | 63.3 |
| LPES (run 1) | 71.4 | 62.2 | 62.0 | 61.0 | 61.6 | 63.6 |
| LPES (run 2) | 71.6 | 62.4 | **62.2** | 60.8 | 61.8 | 63.8 |
| LPES (run 3) | 71.6 | 61.8 | 61.8 | 62.0 | 61.0 | 63.6 |
| LPES (run 4) | **72.6** | 61.0 | 62.0 | **63.2** | 61.0 | **63.9** |
| LPES (run 5) | 72.2 | **62.8** | 61.0 | 61.0 | 60.4 | 63.5 |

Table 8: Performance of LPES across five independent runs compared with baseline methods. Percentages indicate the relative position of relevant documents in the context.

## 5 CONCLUSION

We presented layer-specific positional embedding scaling (LPES), an efficient method to mitigate position bias in transformer-based LLMs. By assigning distinct scaling factors to each layer, LPES achieves a balanced attention distribution across long-context inputs without fine-tuning model parameters or increasing inference latency. To efficiently identify optimal layer-wise scaling factors, we introduced a genetic optimization algorithm constrained by Bézier curves, which significantly reduces the search space and enables rapid convergence with only a few hundred examples. Extensive experiments across multiple benchmarks demonstrate that LPES consistently improves long-context performance while preserving general model capabilities. Notably, LPES requires only a single forward pass, achieving 2.42x speedup over MoICE and 1.45x over Ms-PoE. Our findings also showed that the derived scaling factors generalize well to new tasks and preserve the model's general capabilities, making LPES a broadly applicable and efficient solution.

### ETHICS STATEMENT

This study focuses on positional bias in the contexts of LLMs and strictly adheres to the ICLR Code of Ethics. Ethical considerations were carefully integrated into dataset selection, model usage, methodological design, and potential applications to prevent any involvement of privacy risks, discrimination, or harmful content. All experiments were conducted using publicly available datasets and open-source frameworks to ensure fairness, safety, and reproducibility of the research findings.

### REPRODUCIBILITY STATEMENT

This work ensures strong reproducibility. The datasets, models, and detailed experimental settings are thoroughly described in Section 4.1. The hyperparameters of the genetic algorithm and the weighting scheme of the fitness function are further detailed in Appendices E and G, respectively. All experiments are implemented using the open-source Transformers framework, which further guarantees reproducibility.

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

## A    LONG-TERM DECAY AND ATTENTION WAVE IN ROPE

Zhang et al. (2024) observed that the long-term decay of RoPE causes the model to focus more on the end of a sequence. As the relative distance grows, attention scores drop rapidly, leading the model to overemphasize nearby tokens during autoregressive decoding while neglecting distant ones. To mitigate this issue, they scale RoPE by a factor $s >= 1$ (Figure 3), which effectively reduces the relative distance to $1/s$ of its original value (Figure 4). This adjustment slows the decay rate, enabling the model to attend not only to nearby tokens but also to more distant ones, particularly those in the middle of the sequence.

To demonstrate that scaling RoPE can indeed enhance the model's attention to middle positions, we use the **Vicuna-7B-v1.5** (Chiang et al., 2023) and **LLaMA-2-7B-hf** (Touvron et al., 2023) which both consist of 32 transformer layers to conduct experiment on the validation dataset of **QMSum** (Shaham et al., 2023). We split the context into three parts and calculate the attention scores to the middle-part tokens at different scales. In Figure 5, an increase in the scale factor leads to higher attention scores, demonstrating that scaling RoPE allows the model to focus more on middle-part content during autoregressive decoding.

Chen et al. (2023b) analyze the phenomenon of oscillatory "attention waves" in Transformer models, where attention fluctuates across tokens instead of being smoothly distributed. These oscillations, mainly induced by the mechanisms of RoPE, can cause the model to under-attend to important information located at attention troughs, limiting long-context utilization and potentially introducing instability. To address this issue, the authors propose the *Attention Buckets* approach, which runs multiple model parallels with different bases in RoPE and combines the decoded logits across these bases, producing complementary attention wave patterns. The method enhances the model's sensitivity to context across all positions.

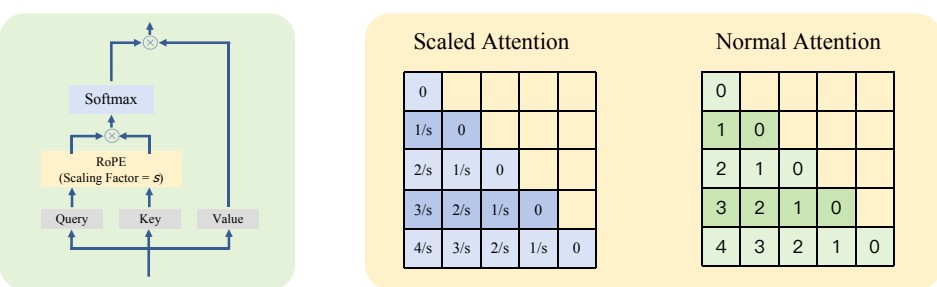

Figure 3: We obtain multi-scale RoPE by scaling the positional indices.

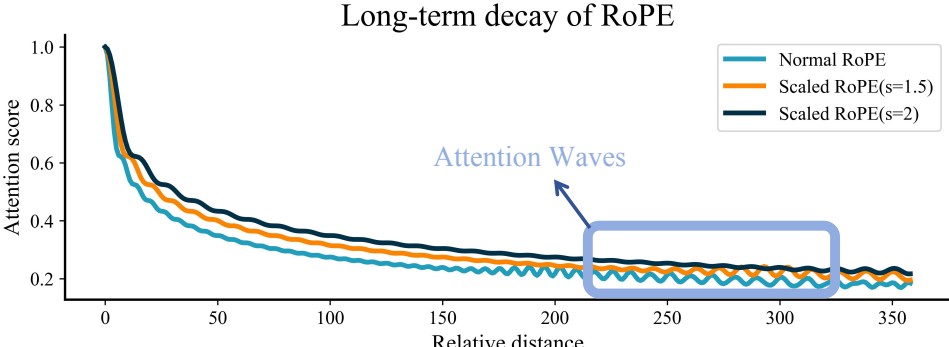

Figure 4: The rapid decay of RoPE prioritizes local focus, and the attention waves may cause the model to overlook crucial information at attention troughs, whereas the scaling operation can slow this decay and generate diverse wave patterns.

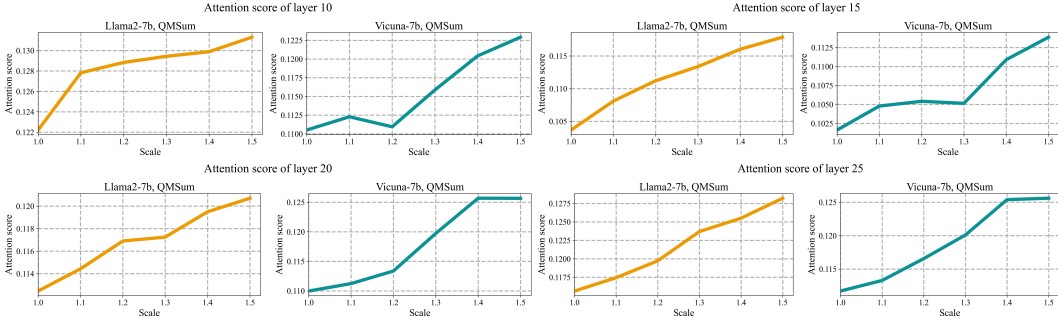

Figure 5: The attention score to the middle part across some layers. The scaling operation can enhance the model's attention to middle positions.

# B    SEARCH SPACE AND TIME COMPLEXITY ANALYSIS

We follow Ding et al. (2024), discretizing the continuous search space to enable more efficient searching. Assume the control points of the Bézier curve are $(P^x, P^y)$, where $P^x \in [0, L-1]$ ($L$ is the number of scaled layers) and $P^y \in [1, 2]$. The values of $P^x$ are discretized with a step size of 1, and the values of $P^y$ are discretized with a step size of $0.1$. Given that the model consists of 32 layers, there are 32 possible selections in $P^X$, while the scaling factor chosen from the $P^Y$ set offers 11 options as shown in Table 9. The total number of choices for the brute-force search is $11^{32}$. If a Cubic Bezier curve is used, each control point has $32 \times 11$ possible combinations. With four control points, the total search space is $352^4$ which approximately narrows the search space by a significant factor $10^{20}$ compared to the brute-force search.

| Coordinate | Search Space |
|:---:|:---:|
| $P^x$ | $\{0, 1, 2, 3, 4, 5, 6, 7, 8, \ldots, n-4, n-3, n-2, n-1\}$ |
| $P^y$ | $\{1.0, 1.1, 1.2, 1.3, 1.4, 1.5, 1.6, 1.7, 1.8, 1.9, 2.0\}$ |

Table 9: Search space for the control point of Bézier curves.

In our method, the dominant cost of the genetic algorithm arises from evaluating the fitness function, which requires running model inference to assess the effectiveness of different scaling factors. In contrast, the computational overhead of other GA operations—such as assignment, mutation, and crossover—is negligible. Using 4×H100 GPUs, we measured the per-epoch time cost of each operation as follows:

| Operation Type | Time (s) |
|:---|:---:|
| Assigning scaling factor from curve | 5.2 |
| Mutation | 4.5 |
| Crossover | 2.3 |
| Computing fitness via model inference | 1167.4 |

Table 10: Measured runtime per epoch of each operation in the genetic algorithm when using 4×H100 GPUs. Model inference dominates the total cost.

Assume the algorithm runs for at most $M$ epochs and generates $N$ new individuals per epoch, and the search uses $S$ samples. Each individual requires three inference runs (placing the correct document at different positions). Thus, the total number of inference calls is $3NMS$. In practice, we perform data-parallel inference using $N_{\text{card}}$ GPUs with batch size $B$, which reduces the effective runtime to $O((3MNS)/(N_{\text{card}} \cdot B))$.

## C LIMITATIONS OF GRADIENT-BASED METHODS

We also attempted to determine the layer-specific scaling factors using gradient descent, but observed poor convergence behaviors. This may also shed light on why LongRoPE (Ding et al., 2024) and LongRoPE2 (Shang et al., 2025) employ genetic algorithms rather than backpropagation to determine the scaling factors across RoPE dimensions. Although the genetic algorithm incurs higher computational overhead compared to directly optimizing hyperparameters via backpropagation, it consistently converges to a more favorable set of scaling parameters. Furthermore, incorporating Bézier curves significantly accelerates the convergence process.

In the **gradient-based method** setting, we construct three datasets from the MDQA, each containing $2,000$ samples in which the correct document is placed at a different position (i.e., first, middle, or last). In each epoch, a total of $2,000$ samples are drawn from these datasets based on the value of $\lambda$ as specified in Section §4.1, where a larger $\lambda$ indicates a higher probability of sampling from the corresponding dataset. For stable training, we use a batch size of 32, a learning rate of $1e-5$, and train the model for a total of 30 epochs.

For the gradient-based method, we observed that even with a large batch size and a small learning rate, the optimization of scaling factors via backpropagation failed to converge. A possible reason is the limited number of trainable parameters (Sun et al., 2025). We evaluated the model at the 30th epoch and found a significant degradation in performance, as shown in Table 11.

| Model | Method | 0% | 25% | 50% | 75% | 100% |
|---|---|---|---|---|---|---|
| Vicuna-7B-v1.5 | Baseline | 70.4 | 58.0 | 55.4 | 55.4 | 60.4 |
| | Gradient-Based | 67.4 | 54.0 | 51.2 | 52.8 | 55.8 |
| Qwen2.5-7B | Baseline | 69.4 | 61.0 | 62.6 | 58.6 | 63.6 |
| | Gradient-Based | 68.7 | 56.6 | 57.6 | 55.8 | 57.8 |

Table 11: Gradient-based methods lead to accuracy degradation in the MDQA dataset.

## D CUBIC BÉZIER CURVE PARAMETERIZATION FOR LAYER ASSIGNMENT

Consider a cubic Bézier curve with four control points:

$$P_0 = (x_0, y_0), \quad P_1 = (x_1, y_1), \quad P_2 = (x_2, y_2), \quad P_3 = (x_3, y_3). \tag{11}$$

where the $x$-coordinates are strictly increasing since Equation 10:

$$x_0 < x_1 < x_2 < x_3. \tag{12}$$

The parametric form of the cubic Bézier curve is

$$\begin{aligned} x(t) &= (1-t)^3 x_0 + 3(1-t)^2 t x_1 + 3(1-t)t^2 x_2 + t^3 x_3, \\ y(t) &= (1-t)^3 y_0 + 3(1-t)^2 t y_1 + 3(1-t)t^2 y_2 + t^3 y_3, \end{aligned} \tag{13}$$

where $t \in [0, 1]$.

Since the $x_i$ are strictly increasing, the function $x(t)$ is typically monotonic. This property allows the use of a binary search over the interval $[0, 1]$ to efficiently find the parameter $t$ corresponding to any given target value $x$, which defines the function $t(x)$.

# E    HYPERPARAMETERS OF THE CONSTRAINED GENETIC ALGORITHM

| Hyperparameter | Value | Description |
|---|---|---|
| Population_size $N_{\mathrm{ps}}$ | 32 | Number of individuals in initial population generation. |
| Parents_size $N_{\mathrm{pa}}$ | 12 | Number of individuals selected as parents. |
| Max_epoch $T$ | 20 | Maximum number of generatios. |
| Mutation_numbers $N_{\mathrm{mu}}$ | 16 | Number of offspring generated through mutation. |
| Crossover_numbers $N_{\mathrm{cr}}$ | 4 | Number of offspring generated through crossover. |
| Max_crossover_try $N_{\mathrm{ct}}$ | 4 | Maximum attempts allowed to produce valid offspring during crossover. |
| $M_{\mathrm{x}}$ | 2 | Perturbation magnitude of the control point's $x$-coordinate ($P^x$). |
| $M_{\mathrm{y}}$ | 0.2 | Perturbation magnitude of the control point's $y$-coordinate ($P^y$). |

Table 12: Hyperparameter settings of the constrained genetic algorithm

# F    DATASET DETAILS

| Dataset | Question Style | Domain | Metric |
|---|---|---|---|
| Coursera | Multiple Choice | Advanced Courses | Accuracy |
| QuALITY | Multiple Choice | Gutenberg | Accuracy |
| TOEFL | Multiple Choice | English Test | Accuracy |
| SFiction | True/False Questions | Scientific Fiction | Accuracy |

Table 13: Overview and evaluation metrics of the sub-datasets in L-Eval.

| Dataset | Description | Metric |
|---|---|---|
| GovReport | Summarization of long reports | ROUGE-1/2/L |
| SummScreenFD | Summarization of TV show episode scripts | ROUGE-1/2/L |
| QMSum | Query-based summarization over meeting transcripts | ROUGE-1/2/L |
| SQuALITY | Question-focused summarization over stories | ROUGE-1/2/L |
| Qasper | Question answering over research papers | F1 |
| NarrativeQA | Question answering about entire books and movie scripts | F1 |
| SpaceDigest | Aggregated sentiment classification over 50 hotel reviews from Space | Exp_similarity |

Table 14: Overview and evaluation metrics of the sub-datasets in ZeroSCROLLS.

Write a high-quality answer for the given question using only the provided search results (some of which might be irrelevant).

{search_results}

Question: {question}
Answer:

Extract the value corresponding to the specified key in the JSON object below.

JSON data:
{formatted_kv_records}

Key: "{key}"
Corresponding value:

Figure 6: Prompt templates used in MDQA and Key-Value Retrieval datasets.

## G   PERFORMANCE VERSUS VALUES OF HYPER-PARAMETERS $\lambda$

In our experiments, we observed that when scaling RoPE, the model tends to improve performance at early positions while neglecting performance at later positions. Consequently, when setting $\lambda$, we favor assigning larger weights to later positions. Here, we define $\langle \lambda_B, \lambda_M, \lambda_E \rangle$ as the weights assigned to the accuracy of the beginning, middle, and end positions, respectively, in the genetic algorithm's fitness function. In this study, we compare three weighting schemes: $\langle 0.333, 0.333, 0.333 \rangle$, $\langle 0.1, 0.3, 0.6 \rangle$, and $\langle 0.2, 0.3, 0.5 \rangle$.

| Method | 0% | 25% | 50% | 75% | 100% | Average |
|---|---|---|---|---|---|---|
| Baseline | 70.4 | 58.0 | 55.4 | 55.4 | 60.4 | 59.9 |
| $\langle 0.333, 0.333, 0.333 \rangle$ | **73.2** | **62.4** | 60.2 | 58.8 | 58.2 | 62.6 |
| $\langle 0.1, 0.3, 0.6 \rangle$ | 70.6 | 60.2 | 60.8 | **61.0** | **62.0** | 63.0 |
| $\langle 0.2, 0.3, 0.5 \rangle$ | 71.4 | 62.2 | **62.0** | **61.0** | 61.6 | **63.2** |

Table 15: The impact of hyper-parameters $\lambda$ on the optimized layer-wise scaling factors, showing that performance is largely insensitive to their choice.

## H   FURTHER ANALYSIS OF CURVE PARAMETERIZATIONS

To further emphasize the advantage of Bézier curves over linear interpolation, we evaluate a deeper model (Vicuna-13B-v1.5 with 40 layers) using finer-grained 10% evaluation intervals. As shown in Table 16, **linear interpolation** exhibits noticeable performance drops around the 20% and 80% positions. In contrast, the smoother **Bézier curve** consistently improves performance across all positions, confirming its superiority in modeling gradual layer-wise variations.

| Curve Type | 0% | 10% | 20% | 30% | 40% | 50% | 60% | 70% | 80% | 90% | 100% | Average |
|---|---|---|---|---|---|---|---|---|---|---|---|---|
| Baseline | 70.4 | 66.2 | 66.4 | 65.2 | 64.8 | 64.0 | 63.0 | 62.0 | 63.4 | 65.2 | 65.2 | 65.1 |
| Linear Interpolation | **71.8** | 66.4 | 64.4 | 65.0 | 64.6 | 64.0 | 64.6 | 63.4 | 62.8 | 65.2 | 65.2 | 65.2 |
| Bézier Curve | **71.8** | **68.4** | **69.4** | **66.6** | **65.2** | **65.2** | **65.0** | **64.0** | **65.8** | 65.2 | 65.2 | **66.5** |

Table 16: Performance across different curve parameterizations, showing consistent improvements of Bézier curves over linear interpolation and baseline.

## I   EFFECTIVENESS OF LPES ON LONGER CONTEXTS

We conduct experiments on Vicuna-1.5-13B and Qwen-2.5-7B under a 16k-token context setting on L-Eval to verify the effectiveness of LPES in long-context scenarios. The decoding length is set to 512 tokens, so the maximum usable context window is limited to 15,872 tokens. As shown in Table 17, the results demonstrate that our method remains effective on larger models and extended context lengths, highlighting its strong scalability and robustness.

| Model | Method | Coursera | QuALITY | TOEFL | SFiction | Average |
|---|---|---|---|---|---|---|
| Vicuna-13B-v1.5-16k | Baseline | 69.6 | 51.4 | 33.3 | 57.1 | 52.9 |
| | MoICE | 67.4 | **55.6** | 35.7 | 52.6 | 52.8 |
| | LPES (Ours) | **70.6** | 54.4 | **36.0** | **59.4** | **55.1** |
| Qwen2.5-7B | Baseline | 59.8 | 66.3 | 76.6 | 71.8 | 68.6 |
| | MoICE | 59.8 | 66.3 | 78.7 | **73.0** | 69.5 |
| | LPES (Ours) | **63.8** | **69.1** | **88.9** | 72.9 | **73.7** |

Table 17: Results on longer-context settings (16k tokens). LPES consistently improves performance over baseline and MoICE on both Vicuna-13B-v1.5-16k and Qwen2.5-7B, demonstrating strong scalability to larger models and longer context windows.

