# OpenReview forum: "Mitigating Position Bias in Transformers via Layer-Specific Positional Embedding Scaling"
_ICLR.cc/2026/Conference — ICLR 2026 Conference Withdrawn Submission_

### Official Review · Reviewer_MV42 · 2025-10-18

**Soundness:** 2
**Presentation:** 3
**Contribution:** 2
**Rating:** 4
**Confidence:** 4

**Summary:**

This paper proposes a novel method to mitigate the position bias problem of LLMs. The method uses genetic algorithm to search layer-wise ROPE scaling factors, and use Bézier curve to reduce the number of points to be searched. Compared to previous methods such as MsPoE, this method achieves better mitigation of position bias, while does not need additional computation in inference.

**Strengths:**

1. Clear structure, and easy to understand
2. There are adequate baselines and datasets in the experiments
3. This method doesn't need to fine-tune the model, and no additional computation is required during inference, which is superior to previous methods.

**Weaknesses:**

1. My main concern is that, this method's main process is to use genetic algorithm to search for scale factors, which is very similar to LongRoPE's [1] search algorithm. Just like the same method is transferred from pre-training dataset to QA dataset, from dimension-wise to layerwise.

[1] Ding, Yiran, et al. "LongRoPE: extending LLM context window beyond 2 million tokens." Proceedings of the 41st International Conference on Machine Learning. 2024.

2. Although using Bezier curves to search for scale factors is an innovative thought, its necessity is not so strong. From section 4.3.1, it seems that the Bezier curve does not have a significant advantage over linear interpolation.

3. The evaluation results of Qwen2.5 only appeared in Experiment 1, not in Experiments 2 and 3. And the other evaluated models are relatively old, and only have a 4k context window (which is too short for long-context problems). So there are doubts about the generalization and practical application value of the method.

4. This method may not applied to models already using LongRoPE or Yarn.

**Questions:**

Why the results of Qwen2.5 are not shown in experiment 2 and 3?

---

> ### Author Response · Authors · 2025-11-20
> **Rebuttal to Reviewer MV42**
>
> ### 1. My main concern is that, this method's main process is to use genetic algorithm to search for scale factors, which is very similar to LongRoPE's [1] search algorithm. Just like the same method is transferred from pre-training dataset to QA dataset, from dimension-wise to layerwise.
>
>
> 1. **Necessity of Genetic Algorithms**
>    Our work **has repeatedly referenced LongRoPE and LongRoPE2**, both of which emphasize the necessity of using genetic algorithms to solve the combinatorial scaling-factor search problem. Directly learning these factors via backpropagation typically leads to unstable or poor convergence, which makes a genetic algorithm a suitable and efficient end-to-end search strategy for this task.
>
> | **Model**         | **Method**       | **0%** | **25%** | **50%** | **75%** | **100%** |
> |-------------------|------------------|--------|---------|---------|---------|----------|
> | Vicuna-7B-v1.5 | Baseline         | 70.4   | 58.0    | 55.4    | 55.4    | 60.4     |
> | Vicuna-7B-v1.5 | Gradient-Based   | 67.4   | 54.0    | 51.2    | 52.8    | 55.8     |
> | Vicuna-7B-v1.5 | Genetic-Based   | 71.4   | 62.2   | 62.0    | 61.0    | 61.6     |
> | Qwen2.5-7B     | Baseline         | 69.4   | 61.0    | 62.6    | 58.6    | 63.6     |
> |    Qwen2.5-7B    | Gradient-Based   | 68.7   | 56.6    | 57.6    | 55.8    | 57.8     |
> |    Qwen2.5-7B    | Genetic-Based   | 69.6   | 64.8    | 69.2    | 63.0    | 65.4     |
>
>
> 2. **From 1D to 2D Constraints**
>    LongRoPE’s genetic algorithm optimizes scaling factors within a 1D constrained space, and these factors directly influence RoPE’s rotation angle. In contrast, our method performs search in a 2D space of Bézier control points. In addition, we introduce novel 2D geometric constraints into both the mutation and crossover operations:
>    - **x-axis:** enforced monotonic increase
>    - **y-axis:** mutations restricted within bounded ranges
>     These constraints ensure curve continuity and stabilize convergence.
>
> 3. **Task-Specific Fitness Function**
>     LongRoPE aims to improve the model’s extrapolation ability, where **perplexity serves as the fitness function** in its genetic algorithm. In contrast, our work targets contextual positional bias, a fundamentally different problem. Based on this objective, we design **a weighted fitness function** that more accurately reflects the relative importance of different positional regions. We additionally explore alternative weighting strategies and provide a detailed analysis in Appendix G.
>
> ---
>
> ### 2. Although using Bezier curves to search for scale factors is an innovative thought, its necessity is not so strong. From section 4.3.1, it seems that the Bezier curve does not have a significant advantage over linear interpolation.
>
> The purpose of introducing **curves** is to **significantly reduce the search space** (by roughly $10^{20}$, as detailed in Appendix B). We experimented with various curve-based parameterizations and found that **Bézier curves** consistently yield the **most stable convergence** and **strongest performance** among all tested options. This empirical advantage is difficult to theoretically explain due to the **black-box nature of the model**.
>
> Several factors may contribute to this observation:
>
> 1. **Modeling flexibility:** Compared with other curve formulations, Bézier curves can represent a wide range of shapes using only a few control points. This allows them to approximate both **linear interpolation** and **step-like behaviors**, providing sufficient flexibility to identify superior combinations of scaling factors.
>
> 2. **Smoothness:** Our ablation studies suggest that smoother curves generally lead to **better convergence and higher performance**, likely because they induce more stable transitions between scaling factors across layers.
>
> We also evaluate on a deeper model (Vicuna-1.5-13B) with finer-grained intervals（10%）. Linear interpolation shows clear performance drops at around the 20% and 80% positions. In contrast, the smooth Bézier curve improves performance at all positions, confirming its advantage in modeling gradual layer-wise changes.
>
> |curve_type | 0% | 10% | 20% | 30% | 40%| 50% | 60% | 70% | 80% | 90% | 100%| Average|
> |  ----  | ----   | ----   | ----   |----   |----   |----   |----   | ----   | ----   |----   |----   |----   |
> |  Baseline  | 70.4   | 66.2   | 66.4  | 65.2  | 64.8 | 64.0  | 63.0   | 62.0   | 63.4   | 65.2   | 65.2   | 65.1  |
> |  Linear Interpolation   | 71.8   |  66.4   | 64.4   | 65.0  | 64.6 | 64.0   | 64.6   | 63.4   | 62.8   | 65.2   | 65.2 | 65.2  |
> |  Bézier Curve    | 71.8   | 68.4   | 69.4   | 66.6   | 65.2   | 65.2  |65.0   | 64.0   | 65.8   | 65.2   | 65.2   | 66.5  |
>
> ---

---

> ### Author Response · Authors · 2025-11-20
> **Rebuttal to Reviewer MV42**
>
> ### 3. The evaluation results of Qwen2.5 only appeared in Experiment 1, not in Experiments 2 and 3. model only have a 4k context window (which is too short for long-context problems). So there are doubts about the generalization and practical application value of the method.
>
> We evaluate LPES on Qwen-2.5-7B across extended context lengths and a broad suite of long-context benchmarks. The results show that LPES consistently delivers strong performance across diverse long-text evaluation settings.
>
> 4k
> |METHOD   | coursera | quality | toefl |  sfiction | Average|
> |  ----  | ----   | ----   | ----   |----   |----  |
> |  Baseline  | 45.5   | 62.4   | 66.0   |60.9  |58.7  |
> |  MOICE  | 48.1   | 64.3   | 67.3   |66.0  |61.4  |
> |  LPES  | 48.5  | 66.4   | 69.3   |66.4  |62.7  |
>
>
> |METHOD | GovRpt| Qasper |SumScrFd| Qmsum |NarrQA |Squality | SpcDgst| average |
> |  ----  | ----   | ----   | ----   |----   |----  |----  |----  |---- |
> |  Baseline  | 24.76   | 22.92   | 14.69   |16.25   |9.78  |14.85  |53.66 |22.42 |
> |  MOICE  | 25.56   | 23.51   | 15.12   | 23.19 | 10.64  | 16.92  |53.81 |24.11 |
> |  LPES  | 27.56   | 23.91   | 16.18   | 23.19 | 11.97  | 14.92  |53.81 |25.51 |
>
>
>
> 16k
> |METHOD   | coursera | quality | toefl |  sfiction | Average|
> |  ----  | ----   | ----   | ----   |----   |----  |
> | BASELINE | 59.8   | 66.3   | 76.6   |71.8  |68.6 |
> |  MOICE  | 59.8   | 66.3   | 78.7   |73.0  |69.5  |
> |  LPES (ours)  | 63.8   | 69.1   | 88.9  |72.9  |73.7  |
>
>
>
> |METHOD | GovRpt| Qasper |SumScrFd| Qmsum |NarrQA |Squality | SpcDgst| average |
> |  ----  | ----   | ----   | ----   |----   |----  |----  |----  |----  |
> | Baseline | 25.17   | 23.33  | 14.23   |17.36   |6.00  |17.60  |51.12  |22.12  |
> |  MOICE  | 38.77   | 18.83  | 15.21   |17.84   |6.14  |18.06  |53.63  |24.07  |
> |  LPES  | 35.77   | 24.63  | 15.91   |20.84   |8.02  |19.06  |53.77  |25.43  |
>
> ---
>
> ### 4. This method may not applied to models already using LongRoPE or Yarn.
>
> By integrating information from multiple RoPE-scaling strategies, our method mitigates both the long-term decay of RoPE and the attention wave phenomenon, thereby alleviating positional bias. Although prior approaches, such as Ms-PoE, MoICE, and Attention Buckets, were proposed to address these issues, they still suffer from two major limitations:
>
> 1. Heavy reliance on hand-crafted heuristics for determining scaling factors.
> 2. Significant inference overhead, as they require multiple forward passes, resulting in substantial latency.
>
> Our work specifically targets these limitations. We follow their experimental settings and evaluation protocols to ensure a fair comparison and demonstrate the effectiveness of our approach.
>
> Due to constraints on computational resources and time, we leave the integration of LPES with more advanced long-context techniques—such as LongRoPE or Yarn—to future work.

---

> > ### Comment · Reviewer_MV42 · 2025-11-21
> >
> > Using Bezier curves changes the search space from 1D to 2D, making the problem more complicated, but gains a improvement which is still not so significant (in my view). So this paper just introduces a dispensable new module on an existing method, and changes the application scenario, making the contribution fair. So I would keep my score.

---

> ### Author Response · Authors · 2025-11-21
> **Rebuttal to Reviewer MV42**
>
> Thank you very much for your timely feedback and insightful suggestions. We sincerely appreciate your careful reading of our work. In the following section, we provide a detailed discussion on the necessity of the 2D constraint as well as the advantages of incorporating Bézier curves.
>
> ##  Additional complexity introduced by the 2D constraint
> 1. **Layer and scaling factors exhibit a two-dimensional dependency.**
>
>     The relationship between layer and scaling factors is inherently two-dimensional, resulting in a very large search space. Introducing Bézier curves is therefore necessary. It is important to note that LongRoPE addresses long-context extrapolation, whereas our work focuses on positional bias. Although both involve RoPE scaling, the underlying problems are different.
>
>     Unlike LongRoPE—where the scaling factor increases with the RoPE dimension based on NTK theory to enhance extrapolation—the optimal scaling pattern in LPES is inherently non-monotonic across layers, as validated by our experiments. This non-linear variation motivates the necessity of performing the search in a 2D layer–scaling space.
>
>
> 2.  **Dimensionality reduction greatly improves search efficiency.**
>
>     In LongRoPE, each GA individual has the similar dimensionality as the number of RoPE rotation angles. In contrast, LPES uses a cubic Bézier curve with only four control points (8 parameters), futher reducing the search space and enabling efficient convergence within practical runtime.
>
> 3.  **The computation required for determining scaling factors is compensated by the inference-time efficiency.**
>
>     Prior methods rely on manually designed heuristics and extensive ablation studies to determine scaling factors, and often require multiple forward passes during inference, substantially increasing latency. In contrast, our method introduces an end-to-end search algorithm that efficiently identifies the optimal scaling factors without introducing any additional inference overhead, while achieving performance superior to previous approaches.
>
> ## Performance Gain
>
> First, linear interpolation **between control points** is also a layer-wise scaling method proposed in our paper. We ultimately chose the Bézier curve for the following reasons:
>
> 1. Determining factors with the Bézier curve requires slightly more time than linear interpolation, but inference complexity remains similar. Moreover, the Bézier-based scaling consistently outperforms the linear-interpolation-based scaling across all datasets.
>
> 2. Consistent performance improvement across positions: As shown in the table below, for scaling factors determined by the Bézier curve, relevant documents achieve performance gains regardless of their position in the context (percentages in the table indicate the correct document position). In some positions (e.g., 20% and 80%), improvements are particularly significant. This is crucial for addressing positional bias.
>
>
> |curve_type | 0% | 10% | 20% | 30% | 40%| 50% | 60% | 70% | 80% | 90% | 100%| Average|
> |  ----  | ----   | ----   | ----   |----   |----   |----   |----   | ----   | ----   |----   |----   |----   |
> |  Baseline  | 70.4   | 66.2   | 66.4  | 65.2  | 64.8 | 64.0  | 63.0   | 62.0   | 63.4   | 65.2   | 65.2   | 65.1  |
> |  Linear Interpolation between control points   | 71.8   |  66.4   | 64.4   | 65.0  | 64.6 | 64.0   | 64.6   | 63.4   | 62.8   | 65.2   | 65.2 | 65.2  |
> |  Bézier Curve  between control points  | 71.8   | 68.4   | 69.4   | 66.6   | 65.2   | 65.2  |65.0   | 64.0   | 65.8   | 65.2   | 65.2   | 66.5  |

---

### Official Review · Reviewer_Ak4b · 2025-10-24

**Soundness:** 2
**Presentation:** 2
**Contribution:** 3
**Rating:** 4
**Confidence:** 3

**Summary:**

This paper presents a layer-specific positional embedding scaling (LPES) method designed to improve the LLM's attention on the information located in the middle of long contexts.
The core contribution of LPES is an efficient search method that uses a genetic algorithm to optimize the control points of a Bézier curve , which in turn defines the scaling factor for each layer of the LLM, significantly reducing the search space.
Experiments on several long-context benchmarks demonstrate that LPES improves accuracy and is faster at inference than baselines like Ms-PoE and MoICE.

**Strengths:**

1. This paper introduces a novel method to solve the non-trivial problem of selecting optimal scaling factors for each layer, which
utilizes Bézier curves to constrain the search space and employs a specially designed genetic algorithm.
2. This  paper focuses on achieving a balanced attention distribution without fine-tuning and without increasing inference cost, which is highly significant and practical for deploying LLMs in real-world, long-context applications.
3. This paper provides a clear and valuable analysis justifying its choice of a genetic algorithm over gradient descent.

**Weaknesses:**

1. The necessity of Bézier curves is not strongly justified. As shown in Table 6, the Bézier curve's performance is only marginally better than the simpler linear interpolation method. This small gain calls into question whether the added complexity of implementing and optimizing Bézier curves is necessary. The authors should provide a more critical analysis of this trade-off.
2. The method's effectiveness hinges on the scaling factors found by the genetic algorithm, which in turn depend on a small search dataset  and a weighted fitness function. However, it is still unclear how sensitive the "optimal" Bézier curve is to the choice of this small search dataset.
3. Formatting errors: A minor but notable issue is the inconsistent bold formatting in the results tables. By convention, the best-performing metric (typically the maximum value) is bolded to help the reader quickly identify the top method. However, this rule is not applied consistently. For example, in Table 2, for the StableBeluga-7B model, the NarrQA column incorrectly bolds the score 9.910(LPES), rather than the maximum value (10.73 for Baseline). This type of error, where non-maximum values are incorrectly bolded, appears in Tables 1, 2, and 3, and may mislead a reader who is scanning the tables.

**Questions:**

1. Is there a stronger theoretical reason, or an experiment on a larger LLM with more layers, that demonstrates a more critical advantage for the non-linear, smooth curve-fitting that Bézier provides over simple linear interpolation? Besides, a fine-grained evaluation—for example, testing at 10% intervals in Table 6 may help to reveal the differences.
2. What happens if you optimize the factors on the key-value retrieval dataset and then apply them to the MDQA benchmark?  Could you clarify how much the "optimal" curve depends on the choice of this search set?
3. The authors should perform a careful review of Tables 1, 2, and 3 to ensure that the bold formatting is corrected and applied consistently, highlighting only the best score in each column for each model-specific comparison.

---

> ### Author Response · Authors · 2025-11-20
> **Rebuttal to Reviewer Ak4b**
>
> ### 1. The necessity of Bézier curves is not strongly justified. As shown in Table 6, the Bézier curve's performance is only marginally better than the simpler linear interpolation method. This small gain calls into question whether the added complexity of implementing and optimizing Bézier curves is necessary. The authors should provide a more critical analysis of this trade-off.
>
> In our method, the dominant cost of the genetic algorithm lies in evaluating the fitness function, which requires model inference. Other GA operations—assignment, mutation, and crossover—are negligible in comparison. Using 4×H100 GPUs, we measure the per-epoch time of each operation:
>
> | Operation Type                           | Time (s) |
> |----------------------------------------- |-----------|
> | Assigning scaling factor from Bézier curve| 5.2      |
> | Assigning scaling factor from Linear curve| 1.4      |
> | Mutation                                  | 4.5      |
> | Crossover                                 | 2.3      |
> | Computing fitness through model inference | 1167.4   |
>
> With four Bézier control points, the search converges within nine epochs and introduces at most 34.2 seconds of overhead. On 4×H100 GPUs, the optimal scaling configuration can be obtained within 2–3 hours, and this additional search cost is negligible relative to the overall computation budget—yet it consistently improves performance. In practice, the time spent determining these scaling factors is more than compensated by the performance gains achieved during inference.
>
> We also evaluate on **a deeper model** (Vicuna-1.5-13B / 40 layers) with **finer-grained intervals（10%)**. Linear interpolation shows clear performance drops at around the 20% and 80% positions. In contrast, the smooth Bézier curve improves performance at all positions, confirming its advantage in modeling gradual layer-wise changes.
>
> |curve_type | 0% | 10% | 20% | 30% | 40%| 50% | 60% | 70% | 80% | 90% | 100%| Average|
> |  ----  | ----   | ----   | ----   |----   |----   |----   |----   | ----   | ----   |----   |----   |----   |
> |  Baseline  | 70.4   | 66.2   | 66.4  | 65.2  | 64.8 | 64.0  | 63.0   | 62.0   | 63.4   | 65.2   | 65.2   | 65.1  |
> |  Linear Interpolation   | 71.8   |  66.4   | 64.4   | 65.0  | 64.6 | 64.0   | 64.6   | 63.4   | 62.8   | 65.2   | 65.2 | 65.2  |
> |  Bézier Curve    | 71.8   | 68.4   | 69.4   | 66.6   | 65.2   | 65.2  |65.0   | 64.0   | 65.8   | 65.2   | 65.2   | 66.5  |
>
> ---
>
> ### 2. The method's effectiveness hinges on the scaling factors found by the genetic algorithm, which in turn depend on a small search dataset and a weighted fitness function. However, it is still unclear how sensitive the "optimal" Bézier curve is to the choice of this small search dataset.
>
>
> We further evaluate robustness on the MDQA dataset using Vicuna-1.5-7B, where we randomly sample 200 training instances as the search set across multiple runs. Across five independent runs, the average performance is 63.68 with a sample variance of 0.027, demonstrating that our method remains highly stable under different search subsets. Overall, our approach consistently outperforms prior methods, highlighting both the stability and robustness of the proposed search algorithm.
>
> |METHOD   | 0% | 25% | 50% | 75% | 100%| Average|
> |  ----  | ----   | ----   | ----   |----   |----   |----   |
> |  Baseline  | 70.4   | 58.0   | 55.4   | 55.4   |60.4   |59.9   |
> |  Attention Buckets  | 72.6   | 61.4   | 60.6   | 60.8  | 59.6   | 63.0   |
> |  Ms-PoE  | 72.6  | 61.4  | 61.8   | 62.0   | 59.0   |63.5  |
> |  MoICE   | 71.6   | 61.2   | 60.6   |60.8   | 62.4  |63.3   |
> |  LPES（**run 1**）  | 71.4   | 62.2   | 62.0   | 61.0   | 61.6   |63.6   |
> |  LPES（**run 2**）  | 71.6   | 62.4   | 62.2   | 60.8   | 61.8   | 63.8 |
> |  LPES（**run 3**）  | 71.6   | 61.8   | 61.8   | 62.0   | 61.0   | 63.6  |
> |  LPES（**run 4**）  | 72.6   | 61.0   | 62.0   | 63.2   | 61.0   | 63.9  |
> |  LPES（**run 5**）  | 72.2   | 62.8   | 61.0   | 61.0   | 60.4   | 63.5  |

---

> ### Author Response · Authors · 2025-11-20
> **Rebuttal to Reviewer Ak4b**
>
> ### 3. Formatting Errors
>
> Thank you for your careful review. We appreciate you pointing out the formatting issues, and we will thoroughly check and correct all related problems in the revised version.
>
> ---
>
> ### 4. Is there a stronger theoretical reason, or an experiment on a larger LLM with more layers, that demonstrates a more critical advantage for the non-linear, smooth curve-fitting that Bézier provides over simple linear interpolation? Besides, a fine-grained evaluation—for example, testing at 10% intervals in Table 6 may help to reveal the differences.
>
> **Reasons**
>
> First, linear interpolation **between control points** is also a layer-wise scaling method proposed in our paper. We ultimately chose the Bézier curve for the following reasons:
>
> 1. Determining factors with the Bézier curve requires slightly more time than linear interpolation, but inference complexity remains similar. Moreover, the Bézier-based scaling consistently outperforms the linear-interpolation-based scaling across all datasets.
>
> 2. Consistent performance improvement across positions: As shown in the table below, for scaling factors determined by the Bézier curve, relevant documents achieve performance gains regardless of their position in the context (percentages in the table indicate the correct document position). In some positions (e.g., 20% and 80%), improvements are particularly significant. This is crucial for addressing positional bias.
>
> Based on the above, the time spent determining these scaling factors is more than compensated by the performance gains achieved during inference.
>
> ---
>
> **Additional Experimental Results**
>
> We also evaluate on a deeper model (Vicuna-1.5-13B) with finer-grained intervals（10%）. Linear interpolation shows clear performance drops at around the 20% and 80% positions. In contrast, the smooth Bézier curve improves performance at all positions, confirming its advantage in modeling gradual layer-wise changes.
>
> |curve_type | 0% | 10% | 20% | 30% | 40%| 50% | 60% | 70% | 80% | 90% | 100%| Average|
> |  ----  | ----   | ----   | ----   |----   |----   |----   |----   | ----   | ----   |----   |----   |----   |
> |  Baseline  | 70.4   | 66.2   | 66.4  | 65.2  | 64.8 | 64.0  | 63.0   | 62.0   | 63.4   | 65.2   | 65.2   | 65.1  |
> |  Linear Interpolation   | 71.8   |  66.4   | 64.4   | 65.0  | 64.6 | 64.0   | 64.6   | 63.4   | 62.8   | 65.2   | 65.2 | 65.2  |
> |  Bézier Curve    | 71.8   | 68.4   | 69.4   | 66.6   | 65.2   | 65.2  |65.0   | 64.0   | 65.8   | 65.2   | 65.2   | 66.5  |
>
> ---
>
> ### 5. What happens if you optimize the factors on the key-value retrieval dataset and then apply them to the MDQA benchmark?
>
> Although the transferred scaling factors show a performance drop, they still effectively mitigate positional bias. This also highlights a key advantage of our approach: the end-to-end search algorithm can automatically determine task-specific scaling-factor combinations, something previous methods are unable to achieve.
>
> |METHOD   | 0% | 25% | 50% | 75% | 100%| Average|
> |  ----  | ----   | ----   | ----   |----   |----   |----   |
> |  Baseline  | 70.4   | 58.0   | 55.4   | 55.4   |60.4   |59.9   |
> |  Positional Interpolation | 71.2   |  59.6  | 58.8  | 56.4   | 56.2 |  60.4   |
> |  Attention Buckets  | 72.6   | 61.4   | 60.6   | 60.8  | 59.6   | 63.0   |
> |  Ms-PoE  | 72.6  | 61.4  | 61.8   | 62.0   | 59.0   |63.5  |
> |  MoICE   | 71.6   | 61.2   | 60.6   |60.8   | 62.4  |63.3   |
> |  LPES（factors from KV）  | 71.4   | 61.4   | 60.0   | 59.8   | 61.4   |62.8  |
> |  LPES（factors from MDQA）  |71.4   | 62.2   | 62.0   | 61.0   | 61.6   |63.6   |

---

> > ### Comment · Reviewer_Ak4b · 2025-11-24
> >
> > - Thank you for your reply. The additional explanations and experimental results regarding the necessity of Bézier curves (especially the 10% interval evaluation on Vicuna-1.5-13B) and the robustness analysis have addressed my concerns.
> >
> > - It is recommended that the new comparative results on the larger model (Vicuna-1.5-13B) be included in the final version, as they provide strong justification for the proposed method. If possible, please update the manuscript (e.g., adding these to the Appendix) during the discussion period to reflect these changes.

---

> ### Author Response · Authors · 2025-11-25
> **Response to Reviewer Ak4b**
>
> Thank you for your positive assessment of our rebuttal. We appreciate your recognition of the additional analyses regarding the necessity of Bézier curves and the robustness evaluation. We have already submitted a revised version that incorporates these analyses along with the corresponding experimental results. In the final version, we will include additional results on Vicuna-1.5-13B. If you have any further questions, please feel free to let us know.

---

> > ### Comment · Reviewer_Ak4b · 2025-11-27
> >
> > I acknowledge the authors' response regarding the formatting inconsistencies in the results tables. However, after reviewing the revised manuscript, I found that this issue has not been fully resolved, i.e., results in Table 2 (StableBeluga-7B row, SpcDgst column), results in Table 3 (LLaMA-2-7B-chat row, SFiction column).
> > This problem raises concerns about the reliability of the reported experimental results. Therefore, I am inclined to maintain my score.

---

### Official Review · Reviewer_o7wH · 2025-10-26

**Soundness:** 2
**Presentation:** 3
**Contribution:** 2
**Rating:** 4
**Confidence:** 4

**Summary:**

The paper proposes LPES, which assigns unique scaling factors to each transformer layer, it uses Bezier curves to simplify the search for optimal factors and a genetic algorithm to find them efficiently, enabling balanced attention without extra inference latency.

**Strengths:**

1. A major advantage is that LPES achieves better performance on MDQA tasks with no extra overhead during inference compared to baselines.
2. The paper is exceptionally clear and well-structured.

**Weaknesses:**

1. The layer-wise assignment of scaling factors in LPES is not a novel concept. While the introduction of the Bézier curve to further constrain the search space is appreciated, it is neither a necessary nor a unique approach, which ultimately weakens the overall novelty of this paper.

2. The authors lack experiments using larger-scale models and larger context windows to fully validate the effectiveness of LPES.

**Questions:**

1. Given that different attention heads are known to possess distinct functional roles, a natural extension would be to assign an individual scaling factor to each head. LPES, however, employs a single, shared scaling factor across all attention heads within a layer. While this choice significantly reduces the search space, it potentially leads to sub-optimal performance. Could the authors provide a more detailed justification for this design choice?

2. While the paper describes the genetic optimization algorithm, providing a performance curve during the optimization process—specifically, showing the performance of each generation on the training data—would significantly strengthen the argument and be more convincing.

---

> ### Author Response · Authors · 2025-11-20
> **Rebuttal to Reviewer o7wH**
>
> ### 1. The layer-wise assignment of scaling factors in LPES is not a novel concept.
>
> To the best of our knowledge, we are the first to address positional bias by integrating multiple RoPE-scaling strategies at the layer level, rather than at the head level or the model level. Prior to this work, we conducted an extensive survey of research on positional bias and found no existing approach that employs a similar layer-wise multi-scaling formulation. Below, we further articulate how our method differs from the most relevant prior studies based on all related work identified.
>
> **Ms-PoE:** Determines factor ranges through extensive ablations and assigns head-specific scales based on positional sensitivity.
>
> **Attention Buckets:** Relies on predefined factor sets and handcrafted search rules, requiring multiple forward passes with different RoPE variants.
>
> **MoICE:** Extends Attention Buckets by fine-tuning head-level scaling weights with task-specific data and performing seven attention computations during inference.
>
> | Method            | Search Algorithm                 | Inference Cost        | Scaling Level |
> |-------------------|----------------------------------|------------------------|----------------|
> | Ms-PoE                | Manual ablation                  | 2× attention forward   | Head           |
> | MoICE                  | Manual sets and rules + task-specific tuning | 7× attention forward| Head           |
> | Attention Buckets | Manual sets and rules   | 7× model forward     | Model          |
> | LPES (ours)           | **End-to-end search**       | **1× model forward**            | **Layer**          |
>
> ---
>
> ### 2. While the introduction of the Bézier curve to further constrain the search space is appreciated, it is neither a necessary nor a unique approach, which ultimately weakens the overall novelty of this paper.
>
> Determining factors is inherently a combinatorial optimization task, and introducing a Bézier-curve constraint is both necessary and effective: it substantially reduces the search space while preserving sufficient expressiveness for identifying high-quality scaling configurations. Moreover, using Bézier curves to determine layer-wise RoPE scaling factors that alleviate positional bias is, to the best of our knowledge, novel and has not been explored in prior work.
>
>
> We conducted ablations without Bézier modeling (Table 7 in the paper). When Bézier curves are removed (Brute-Force), the search space expands dramatically—as analyzed in Appendix B—resulting in unstable convergence and degraded performance (with a maximum of 20 epochs).
>
> |Number of Control Point  | 0% | 25% | 50% | 75% | 100%| Average| converge epoch |
> |  ----  | ----   | ----   | ----   |----   |----   |----   |----   |
> |  Baseline  | 70.4   | 58.0   | 55.4   | 55.4   |60.4   |59.9   |-
> |  Brute-Force  | 71.8   | 58.2   | 57.2  | 55.6  | 58.0   | 60.2   |20
> |  Bezier curve   | 71.4   | 62.4   | 62.4   |61.4   | 61.4  |63.6   |9
>
> ---
> ### 3. Could the authors provide a more detailed justification for adopting layer-level scaling rather than head-level scaling?
>
> Prior RoPE-scaling approaches—including Ms-PoE, MoICE, and Attention Buckets—require substantial manual tuning, motivating our shift toward a more principled, end-to-end search framework，layer-wise assignment of scaling factors makes end-to-end search both tractable and computationally efficient.
>
> Determining scaling factors is fundamentally a combinatorial optimization problem. While genetic algorithms provide an effective optimization mechanism, conducting the search at the head level results in an enormous search space, slow convergence, and prohibitive computational cost. To address this, we move the search from the head level to the layer level, which reduces the search space by orders of magnitude. This makes end-to-end search both tractable and computationally efficient.
>
> For a model with $L$ layers and $H$ attention heads per layer, and $N$ candidate scaling values, the search space sizes differ significantly:
>
> - **Head-wise search:** $N^{L \times H}$
> - **Layer-wise search:** $N^{L}$
> - **Reduction:** $N^{L \times (H-1)}$
>
> This exponential reduction makes the search space substantially more tractable. To validate this, we conducted **a brute-force genetic search** at both the layer level and the head level using the same genetic algorithm setup. The results (Vicuna-1.5-7B on MDQA, with a maximum of 20 epochs) are presented below.
>
>
> | Method | 0% | 25% | 50% | 75% | 100%| Average|
> |  ----  | ----   | ----   | ----   |----   |----   |---- |
> |  Baseline  | 70.4   | 58.0   | 55.4   | 55.4   |60.4   |59.9
> |  head-level genetic algorithm  | 68.8   | 55.2   | 53.8  | 53.0  | 56.4   | 57.4
> |  layer-level genetic algorithm | 71.4   | 56.4   | 58.6   | 55.6  | 59.2   | 60.2   |
>
> These results confirm that layer-level scaling provides a much more favorable search landscape and significantly better performance.

---

> ### Author Response · Authors · 2025-11-20
> **Rebuttal to Reviewer o7wH**
>
> ### 4. The authors lack experiments using larger-scale models and larger context windows to fully validate the effectiveness of LPES.
>
> Due to current computational and time constraints, we conduct experiments on Vicuna-1.5-13B and Qwen-2.5-7B under a 16k-token context setting. The results show that our method remains effective on larger models and longer context lengths, demonstrating strong scalability and robustness.
>
> vicuna-13b-v1.5-16k
> |METHOD   | coursera | quality | toefl |  sfiction | Average|
> |  ----  | ----   | ----   | ----   |----   |----  |
> | BASELINE | 69.6   | 51.4   | 33.3    |57.1  |52.9 |
> |  MOICE   | 67.4   | 55.6   | 35.7    |52.6  |52.8  |
> |  LPES (ours)   | 70.6   | 54.4   | 36.0    |59.4  |55.1  |
>
> qwen2.5-7b
> |METHOD   | coursera | quality | toefl |  sfiction | Average|
> |  ----  | ----   | ----   | ----   |----   |----  |
> | BASELINE | 59.8   | 66.3   | 76.6   |71.8  |68.6 |
> |  MOICE  | 59.8   | 66.3   | 78.7   |73.0  |69.5  |
> |  LPES (ours)  | 63.8   | 69.1   | 88.9  |72.9  |73.7  |
>
> ---
>
> ### 5. While the paper describes the genetic optimization algorithm, providing a performance curve during the optimization process—specifically, showing the performance of each generation on the training data—would significantly strengthen the argument and be more convincing.
>
>
> Table 7 reports the performance impact of using Bezier curves with different numbers of control points. It does **not** track performance changes during the optimization process. We will revise the table structure to make this distinction clearer.
>
> |Number of Control Point  | 0% | 25% | 50% | 75% | 100%| Average|
> |  ----  | ----   | ----   | ----   |----   |----   |----   |
> |  Baseline  | 70.4   | 58.0   | 55.4   | 55.4   |60.4   |59.9   |
> |  2  | 70.8   | 59.4   | 57.6   | 55.6  | 59.8   | 60.6   |
> |  3  | 71.8  | 60.2  | 59.2   | 59.9   | 60.0   |62.2  |
> |  4  | 71.4   | 62.2   | 62.0   | 61.0   | 61.6   |63.6   |
> |  5   | 71.4   | 62.4   | 62.4   |61.4   | 61.4  |63.8   |

---

> ### Comment · Reviewer_o7wH · 2025-11-24
>
> Your previous response did not adequately address my concerns. I still have the following unresolved questions:
>
> 1.  The `head-wise search` strategy inherently possesses a larger search space. Despite this, the authors still employ a `brute-force genetic search` as the search strategy. I hypothesize that the suboptimal choice of search strategy is responsible for the performance degradation. Also, could the authors elaborate, from the perspective of the model mechanism or other relevant angles, on why the `Layer-wise search`, which offers lower degrees of freedom, yields superior results?
>
> 2.  I am particularly curious about the substantial performance difference observed across various base models:
>    ` LPES with Qwen2.5-7B` achieved a 10-point improvement over the baseline on the TOEFL benchmark.
>     `LPES with Vicuna-13B-v1.5-16K` showed only a 1-point improvement. I find this disparity in results quite perplexing.
>
> 3.  The authors failed to provide a performance curve during the optimization process. Furthermore, they did not offer any explanation as to why this crucial information could not be presented.

---

> ### Author Response · Authors · 2025-11-26
> **Response to Reviewer o7wH**
>
> We sincerely appreciate your prompt response. Your comments have prompted further reflection and helped us improve our manuscript. We will address your concerns in the following sections.
>
> ---
>
> ### 1. The head-wise search strategy inherently possesses a larger search space. Despite this, the authors still employ a brute-force genetic search as the search strategy. I hypothesize that the suboptimal choice of search strategy is responsible for the performance degradation.
>
>
> Because the mapping between attention heads and their optimal scaling factors is highly irregular and lacks smooth structural patterns, it cannot be effectively captured using curve-based parameterizations such as Bézier curves. Therefore, we compare brute-force genetic search at both the layer level and the head level. Under the same search budget (20 epochs), layer-level search operates in a substantially smaller and more manageable space, allowing the algorithm to converge quickly to high-quality scaling factors. In contrast, head-level search faces a much larger and more complex search space, leading to slower convergence and less stable optimization. **While exploring scaling factors at the head level may potentially yield better performance, it also introduces a substantially larger search space, leading to significant time and computational overhead and limiting the practical applicability of the method.**
>
> We greatly appreciate your insightful comments. We plan to further investigate the application of genetic algorithms for head-level scaling factor optimization in future work.
>
>
> ---
>
>
> ### 2. could the authors elaborate, from the perspective of the model mechanism or other relevant angles, on why the Layer-wise search, which offers lower degrees of freedom, yields superior results?
>
> By constraining the search space and reducing the degrees of freedom, the algorithm can converge to a high-quality set **within a limited time**, even if minor performance trade-offs occur, thereby improving practical applicability. Similarly, LongRoPE [1] employs constraints to narrow the search space, enabling the genetic algorithm to converge efficiently despite the reduced degrees of freedom.
>
> **Under limited computational resources,** head-level search struggles due to the larger and more irregular search space and the lack of a smooth mapping between attention heads and their optimal scaling factors. In contrast, performing the search at the layer level with Bézier curve constraints drastically reduces the search space, enabling the genetic algorithm to converge rapidly—within 9 epochs—to a high-quality set of scaling factors.
>
>
>
> [1] Ding Y, Zhang L L, Zhang C, et al. LongRoPE: Extending LLM Context Window Beyond 2 Million Tokens[C]//International Conference on Machine Learning. PMLR, 2024: 11091-11104.

---

> ### Author Response · Authors · 2025-11-26
> **Response to Reviewer o7wH**
>
> ### 3. I am particularly curious about the substantial performance difference observed across various base models: LPES with Qwen2.5-7B achieved a 10-point improvement over the baseline on the TOEFL benchmark. LPES with Vicuna-13B-v1.5-16K showed only a 1-point improvement. I find this disparity in results quite perplexing.
>
> We further examined the results, and the observed disparity may be attributed to differences in the underlying mechanisms of the respective base models. In the “lost-in-the-middle” experiment on Qwen2.5-7B (Table 1), LPES shows clear improvements at middle context positions—particularly at the 50% position—indicating the method remains effective across models, though the magnitude of gains may vary.
>
> ### 4. The authors failed to provide a performance curve during the optimization process. Furthermore, they did not offer any explanation as to why this crucial information could not be presented.
>
> Thank you for your suggestion. We used 200 samples to determine the scaling factors for Vicuna-7B-v1.5 and present the performance on both the search set and the test set across the search epochs. To better illustrate the trends, we report performance at epochs 1, 3, 5, 7, 9, 11, and 13.
>
> As shown in the table below, the fitness of the best individual steadily increases on the search set as the genetic algorithm progresses, reaching convergence after 9 epochs. The fitness function assigns weights of 0.2, 0.3, and 0.5 at the 0%, 50%, and 100% positions, respectively.
>
>
> |Epoch| 0% | 50% | 100% | Fitness |
> |---  | ---| ----| ---- |----     |
> |1  | 76.0 | 61.7 | 65.5 |66.4    |
> |3  | 78.0 | 62.5 | 67.0 |67.9    |
> |5  | 77.0 | 64.5 | 67.5 |68.5    |
> |7  | 77.0 | 66.0 | 68.0 |69.2    |
> |9  | 77.5 | 66.5 | 68.0 |69.5    |
> |11 | 77.5 | 66.5 | 68.0 |69.5    |
> |13 | 77.5 | 66.5 | 68.0 |69.5    |
>
> The performance on the test set (Table below), which contains 500 samples, also shows steady improvement throughout the search.
>
> |Method| 0% | 25% | 50% | 75% | 100%               |Average|
> |  ---| ----   | ----   | ----   |----    |----    |----   |
> |  baseline | 70.4   | 58.0   | 55.4   | 55.4   | 60.4   |59.9   |
> |  epoch 1  | 70.6   | 60.8   | 60.0   | 57.2   | 59.2   |61.6   |
> |  epoch 3  | 74.0   | 59.6   | 58.2   | 61.4   | 60.0   |62.6   |
> |  epoch 5  | 71.0   | 60.8   | 61.0   | 60.2   | 60.6   |62.7   |
> |  epoch 7  | 71.4   | 61.8   | 61.8   | 60.8   | 61.0   |63.4   |
> |  epoch 9  | 71.4   | 62.2   | 62.0   | 61.0   | 61.6   |63.6   |
> |  epoch 11 | 71.4   | 62.2   | 62.0   | 61.0   | 61.6   |63.6   |
> |  epoch 13 | 71.4   | 62.2   | 62.0   | 61.0   | 61.6   |63.6   |
>
>
> We will include this performance curve in the paper to give readers a clearer understanding of the optimization process.
>
> ---
> We sincerely thank you again for your valuable suggestions. If you have any further questions, please do not hesitate to let us know, and we will respond promptly.

---

### Official Review · Reviewer_oYqR · 2025-10-31

**Soundness:** 2
**Presentation:** 3
**Contribution:** 2
**Rating:** 4
**Confidence:** 3

**Summary:**

This paper aims to address the “lost-in-the-middle” problem by assigning a distinct rotary position embedding (RoPE) scaling factor to each layer. While previous works combine the output from multiple inferences with different scaling factors, the proposed method is claimed to require inference only once. In determining the scaling factor for each layer, this paper proposes a genetic algorithm to adjust Bezier curves corresponding to different scaling factor settings.

**Strengths:**

* The proposed method seems straightforward. The empirical evidence presented demonstrates that when the scaling factor is appropriately set, the proposed method can achieve good results with a single forward inference.

* The paper is clear and easy to follow; the proposed method is clearly demonstrated.

* The design of the genetic algorithm for the scaling factor setting is interesting and creative. Generally, the perspective of searching and adjusting the scaling factor provides a brand new perspective.

**Weaknesses:**

* The time complexity of the algorithm is of primary concern. Despite this paper emphasizing that previous methods require multiple forward passes, the proposed method requires first determining the scaling factor. The proposed genetic algorithm involves inference on an additional dataset and gradually adjusting the scaling factor for each layer, which seems to require extensive computation.

* The extensive efforts devoted to adjusting the scaling factor of each layer further raise my concern about the robustness of the proposed method, where the good performance may highly rely on an appropriate scaling factor setting and may be limited when there is no high-quality dataset to apply the genetic algorithm on. In Tables 2 and 3, the improvement of the proposed method over the previous work, MoICE, appears marginal.

* While the paper provides results showing that Bezier curves achieve better performance than the step function and linear interpolation, the choice of the curve type could use more insights and reasons to explain why Bezier curves are a good choice.

**Questions:**

My questions are generally related to the concerns I mentioned above.

* Could the authors provide a time complexity analysis of the proposed genetic algorithm? Running the genetic algorithm for a dozen epochs seems computationally expensive.

 * More analyses on how the scaling factor would affect the model's performance would be appreciated. Since it is the core contribution of the proposed method, the current analyses on the scaling factor do not appear to be in-depth. Why are Bezier curves a good choice for scaling factor setting? Are there any insights that could inform the setting of the scaling factor?

* While the scaling factor of RoPE is the focus of the proposed method and many previous methods, could the proposed genetic algorithm be applied to other methods that involve changing the scaling factor?

---

> ### Author Response · Authors · 2025-11-20
> **Rebuttal to Reviewer oYqR**
>
> We appreciate your careful evaluation and insightful comments, and we address the concerns in detail below.
>
> ### 1. The time complexity of the algorithm is of primary concern. Despite this paper emphasizing that previous methods require multiple forward passes, the proposed method requires first determining the scaling factor.
>
>
> Previous methods that mitigate positional bias through RoPE scaling—such as Ms-PoE, Attention Buckets, and MoICE—**all require determining scaling factors via compute-intensive search procedures.** Thus, performing an initial scaling-factor search is fundamentally unavoidable.
>
> Prior methods rely on manually designed search spaces and handcrafted rules (detailed below). In contrast, we introduce a fully end-to-end search algorithm that generalizes across architectures and tasks.
>
>
> Details on the Search Procedures of Prior Work:
>
> 1. **Ms-PoE**
>    Determines scaling factors through extensive ablation studies on the validation set.
> 2. **Attention Buckets**
>    Relies on predefined factor sets together with handcrafted search rules.
> 3. **MoICE**
>    Builds on Attention Buckets and further incorporates dataset-specific weighting through fine-tuning.
>
> These approaches depend heavily on **manual tuning** and **model-specific design choices**, which prevent them from capturing the model’s intrinsic contextual processing behavior.
>
> ---
>
> ### 2. The proposed genetic algorithm involves inference on an additional dataset and gradually adjusting the scaling factor for each layer, which seems to require extensive computation.
>
> Selecting suitable scaling factors is fundamentally a combinatorial optimization problem. Directly optimizing these factors through backpropagation often leads to unstable training dynamics and suboptimal solutions (as reflected by the MDQA results in the table below). In contrast, genetic algorithms provide a more stable and effective end-to-end search strategy for this task. In practice, the time required to identify high-quality scaling factors is easily justified by the substantial performance gains achieved during inference.
>
> | **Model**         | **Method**       | **0%** | **25%** | **50%** | **75%** | **100%** |
> |-------------------|------------------|--------|---------|---------|---------|----------|
> | Vicuna-7B-v1.5 | Baseline         | 70.4   | 58.0    | 55.4    | 55.4    | 60.4     |
> | Vicuna-7B-v1.5 | Gradient-Based   | 67.4   | 54.0    | 51.2    | 52.8    | 55.8     |
> | Vicuna-7B-v1.5 | Genetic-Based   | 71.4   | 62.2   | 62.0    | 61.0    | 61.6     |
> | Qwen2.5-7B     | Baseline         | 69.4   | 61.0    | 62.6    | 58.6    | 63.6     |
> |    Qwen2.5-7B    | Gradient-Based   | 68.7   | 56.6    | 57.6    | 55.8    | 57.8     |
> |    Qwen2.5-7B    | Genetic-Based   | 69.6   | 64.8    | 69.2    | 63.0    | 65.4     |
>
> To mitigate the potentially large search space and high computational cost of genetic algorithms, we introduce a Bézier-curve–based constraint that significantly reduces the search space while maintaining expressiveness. This design allows the algorithm to efficiently converge to high-quality scaling-factor combinations within a limited time (results are shown in the table below, evaluated on Vicuna-1.5-7B using MDQA, with a maximum of 20 epochs).
>
> | Curve Type     | Average Performance | Convergence Epoch |
> |----------------|------------------|-----------------|
> | Baseline     | 59.9             | -            |
> | Brute-Force    | 60.2             | 20              |
> | Bézier Curve   | 63.6             | 9               |
>
> Under the default experimental setup in our paper（Appendix E）, using 4×H100 GPUs, the search procedure identifies strong scaling-factor candidates within **2–3 hours**, demonstrating both practicality and scalability.

---

> ### Author Response · Authors · 2025-11-20
> **Rebuttal to Reviewer oYqR**
>
> ### 3. Could the authors provide a time complexity analysis of the proposed genetic algorithm?
>
> In our method, the dominant cost of the genetic algorithm comes from evaluating the fitness function, which requires running model inference to assess the effectiveness of different scaling factors. In contrast, the computational cost of other GA operations—such as assignment, mutation, and crossover—is negligible. Using 4×H100 GPUs, we measured the time spent on each operation per epoch as follows:
>
> | Operation Type                           | Time (s) |
> |----------------------------------------- |-----------|
> | Assigning scaling factor from curve      | 5.2       |
> | Mutation                                 | 4.5       |
> | Crossover                                | 2.3       |
> | Computing fitness through model inference| 1167.4     |
>
>
> Assuming the algorithm runs for at most $M$ epochs and generates $N$ new individuals per epoch, and the search uses $S$ samples, each individual requires three inference runs (placing the correct document at different positions). Thus, the total number of inference calls is $3 N M S$. In practice, we perform data-parallel inference on $N_{\text{card}}$ GPUs with batch size $B$, reducing the effective runtime to $O\left(3 M N S / (N_{\text{card}} \cdot B\right)).$  With 4×H100 GPUs, the optimal scaling factors can be obtained within **2–3 hours**.
>
> ---
>
> ### 4. The extensive efforts devoted to adjusting the scaling factor of each layer further raise my concern about the robustness of the proposed method where the good performance may highly rely on an appropriate scaling factor setting.
>
> We further evaluate robustness on the MDQA dataset using Vicuna-1.5-7B, where we randomly sample 200 training instances as the search set across multiple runs. Across five independent runs, the average performance is 63.68 with a sample variance of 0.027, demonstrating that our method remains highly stable under different search subsets. Overall, our approach consistently outperforms prior methods, highlighting both the stability and robustness of the proposed search algorithm.
>
> |METHOD   | 0% | 25% | 50% | 75% | 100%| Average|
> |  ----  | ----   | ----   | ----   |----   |----   |----   |
> |  Baseline  | 70.4   | 58.0   | 55.4   | 55.4   |60.4   |59.9   |
> |  Attention Buckets  | 72.6   | 61.4   | 60.6   | 60.8  | 59.6   | 63.0   |
> |  Ms-PoE  | 72.6  | 61.4  | 61.8   | 62.0   | 59.0   |63.5  |
> |  MoICE   | 71.6   | 61.2   | 60.6   |60.8   | 62.4  |63.3   |
> |  LPES（**run 1**）  | 71.4   | 62.2   | 62.0   | 61.0   | 61.6   |63.6   |
> |  LPES（**run 2**）  | 71.6   | 62.4   | 62.2   | 60.8   | 61.8   | 63.8 |
> |  LPES（**run 3**）  | 71.6   | 61.8   | 61.8   | 62.0   | 61.0   | 63.6  |
> |  LPES（**run 4**）  | 72.6   | 61.0   | 62.0   | 63.2   | 61.0   | 63.9  |
> |  LPES（**run 5**）  | 72.2   | 62.8   | 61.0   | 61.0   | 60.4   | 63.5  |
>
> ---
>
> ### 5. May be limited when there is no high-quality dataset to apply the genetic algorithm on.
>
> Our method exhibits strong robustness: even when the search set varies across runs, the algorithm consistently converges to stable, high-performing configurations of scaling factors. As shown in the results above, modifying the search subset leads to only minimal performance variation. Furthermore, the scaling factors identified from the MDQA search generalize well across both models and tasks. They transfer effectively to open-ended tasks on Zero-Scroll as well as closed-ended tasks on LEVAL (Tables 2–3).

---

> ### Author Response · Authors · 2025-11-20
> **Rebuttal to Reviewer oYqR**
>
> ### 6. In Tables 2 and 3, the improvement of the proposed method over the previous work, MoICE, appears marginal.
>
> MoICE relies on scaling factors obtained from a RoPE-only search algorithm whose search space and search rules must be manually defined. **It further requires an additional dataset to fine-tune head-level scaling weights** and performs attention computations seven times, leading to substantial memory and latency overhead.
>
>
> In contrast, our method employs an end-to-end search algorithm that simultaneously accounts for both contextual processing and the RoPE mechanism, requires no long-context fine-tuning, and introduces no additional computational or memory cost in inference, **while achieving comparable or better performance.**
>
> | Attribute        | MoICE                                                                     | LPES                                                                 |
> | ---------------- | ------------------------------------------------------------------------- | -------------------------------------------------------------------- |
> | Search Algorithm | Considers only RoPE waveform                                              | End-to-end; considers both RoPE and contextual processing mechanisms |
> | Fine-tuning      | Requires long-context fine-tuning                                         | Not required                                                         |
> | Inference        | Multiple attention computations increase latency (2.42×) and memory usage | No additional computational or memory overhead                       |
>
> ---
> ### 7. While the paper provides results showing that Bezier curves achieve better performance than the step function and linear interpolation, the choice of the curve type could use more insights and reasons to explain why Bezier curves are a good choice.
>
> The purpose of introducing curves is to **significantly reduce the search space** (by roughly $10^{20}$, as detailed in Appendix B).  Linear interpolation **between control points** is also a layer-wise scaling method proposed in our paper. We ultimately chose the Bézier curve for the following reasons:
>
> 1. Determining factors with the Bézier curve requires slightly more time than linear interpolation, but inference complexity remains similar. Moreover, the Bézier-based scaling consistently outperforms the linear-interpolation-based scaling across all datasets.
>
> 2. Consistent performance improvement across positions: As shown in the table below, for scaling factors determined by the Bézier curve, relevant documents achieve performance gains regardless of their position in the context (percentages in the table indicate the correct document position). In some positions (e.g., 20% and 80%), improvements are particularly significant. This is crucial for addressing positional bias.
>
> Based on the above, the time spent determining these scaling factors is more than compensated by the performance gains achieved during inference.
>
> We evaluate on **a deeper model** (Vicuna-1.5-13B / 40 layers) with **finer-grained intervals（10%）**. Linear interpolation shows clear performance drops at around the 20% and 80% positions. In contrast, the smooth Bézier curve improves performance at all positions, confirming its advantage in modeling gradual layer-wise changes.
>
> |curve_type | 0% | 10% | 20% | 30% | 40%| 50% | 60% | 70% | 80% | 90% | 100%| Average|
> |  ----  | ----   | ----   | ----   |----   |----   |----   |----   | ----   | ----   |----   |----   |----   |
> |  Baseline  | 70.4   | 66.2   | 66.4  | 65.2  | 64.8 | 64.0  | 63.0   | 62.0   | 63.4   | 65.2   | 65.2   | 65.1  |
> |  Linear Interpolation   | 71.8   |  66.4   | 64.4   | 65.0  | 64.6 | 64.0   | 64.6   | 63.4   | 62.8   | 65.2   | 65.2 | 65.2  |
> |  Bézier Curve    | 71.8   | 68.4   | 69.4   | 66.6   | 65.2   | 65.2  |65.0   | 64.0   | 65.8   | 65.2   | 65.2   | 66.5  |

---

> ### Author Response · Authors · 2025-11-20
> **Rebuttal to Reviewer oYqR**
>
> ### 8. More analyses on how the scaling factor would affect the model's performance would be appreciated.
> In Appendix A, we provide a detailed analysis showing that:
>
> 1. The long-range decay property of RoPE makes the model over-focus on nearby tokens while neglecting information in the middle of long contexts.
> 2. Different RoPE scaling factors exhibit distinct attention peaks and valleys, which introduce positional bias.
>
> Combining multiple RoPE scalings helps alleviate long-term decay and the attention-wave phenomenon, thereby reducing positional bias. Although prior methods are based on similar principles, they suffer from two major limitations: they require multiple forward passes during the search process and depend heavily on manual heuristics to determine the scaling factors. Our method addresses both issues. Furthermore, our experiments (Figure 5) show that the scaling operation increases the model’s sensitivity to mid-context tokens across layers.
>
> ---
>
> ### 9. While the scaling factor of RoPE is the focus of the proposed method and many previous methods, could the proposed genetic algorithm be applied to other methods that involve changing the scaling factor?
>
> The determination of scaling factors in prior approaches can also be viewed as a combinatorial optimization problem, for which genetic algorithms are generally recognized as an effective solution. In principle, it is feasible to further optimize these earlier methods. However, directly applying a brute-force genetic search may lead to suboptimal solutions due to the extremely large search space.
>
> However, one of the key issues our method addresses is the need for multiple forward passes during inference—an overhead that these methods inherently suffer from. Since our approach eliminates this bottleneck, we do not further explore applying genetic algorithms to those earlier techniques.

---

> > ### Comment · Reviewer_oYqR · 2025-11-24
> > **Thank you for the response**
> >
> > The authors have provided a detailed rebuttal that addresses most of my concerns. I intend to raise my rating accordingly.
> > There are just a few additional questions:
> >
> > * Since I am not particularly familiar with the baseline methods, could the authors provide a direct comparison of the runtime overhead between previous methods and the proposed method? (Add some runtime results of baseline methods mentioned in the paper)
> >
> > * I appreciate the results of adjusting factors on the randomly sampled search set. I wonder whether the factor can be generalized to other datasets? (It is a question out of my curiosity, which would not affect my rating.)

---

> > > ### Author Response · Authors · 2025-11-24
> > > **Response to Reviewer oYqR**
> > >
> > > Thank you very much for your timely feedback and insightful suggestions, which have prompted us to reflect more deeply and helped improve the quality of our paper. We provide detailed responses to your questions below.
> > >
> > > ---
> > >
> > > ### 1. Could the authors provide a direct comparison of the runtime overhead between previous methods and the proposed method?
> > >
> > > To highlight the inference efficiency of our approach, we randomly sampled 500 examples from the MDQA dataset and measured the average inference time of Vicuna on a single H100 GPU (batch size = 1). For a fair comparison, all methods used FlashAttention-2 as the attention backend. **As summarized in the table below (Table 5 in our paper), the average inference time per sample is 1.03 seconds for Ms-PoE, 1.72 seconds for MoICE, and approximately 0.71 seconds for our method. These results indicate that LPES is roughly 1.45× faster than Ms-PoE and 2.42× faster than MoICE.**
> > >
> > > | Method             | Inference time per sample (s) |
> > > |-------------------|-------------------------------|
> > > | Baseline           | 0.71                          |
> > > | Attention Buckets  | 3.38                          |
> > > | Ms-PoE             | 1.03                          |
> > > | MoICE              | 1.72                          |
> > > | LPES (Ours)        | **0.71**                      |
> > >
> > > A more detailed explanation can be found in Section 4.2 (Results) of our paper. We will consider refining this section to present the analysis more clearly and prominently.
> > >
> > > ---
> > >
> > > ### 2. I appreciate the results of adjusting factors on the randomly sampled search set. I wonder whether the factor can be generalized to other datasets? (It is a question out of my curiosity, which would not affect my rating.)
> > >
> > > Your question aligns with one of the key aspects we intended to investigate. To assess the cross-dataset generalization of the scaling factors, we applied the factors learned from the MDQA search set to the ZeroSCROLLS and L-Eval benchmarks. As reported in Tables 2 and 3 of our paper, these factors consistently enhance performance on long-context tasks, demonstrating strong generalization capability across different datasets.
> > >
> > > ---
> > >
> > >
> > > Please do not hesitate to let us know if you have any further questions; we would be glad to respond promptly.

---

### Comment · Area_Chair_77fZ · 2025-11-27

Dear reviewers,

A reminder that the discussion phase will end in a few days (**December 2**). Engaging with the author's rebuttal is essential to address all potential concerns before our final discussion stage.

Thanks,
The AC

---

### Note · Authors · 2026-01-06

**Comment:**

Dear Reviewers,

We sincerely appreciate the time and effort you have dedicated to reviewing our submission. Your valuable feedback has provided us with significant insights, which will be invaluable as we continue to refine our work.

After careful consideration, we have decided to withdraw our paper from the conference. We believe this decision will allow us to further improve the quality of our research before presenting it to the community.

Thank you once again for your understanding and support.

**Withdrawal Confirmation:**

I have read and agree with the venue's withdrawal policy on behalf of myself and my co-authors.